# Comprehensive innate immune profiling of chikungunya virus infection in pediatric cases

Daniela Michlmayr[1,†], Theodore R Pak[2,†], Adeeb H Rahman[2,3], El-Ad David Amir[2,3], Eun-Young Kim[4], Seunghee Kim-Schulze[2,3], Maria Suprun[5], Michael G Stewart[4], Guajira P Thomas[4], Angel Balmaseda[6], Li Wang[2], Jun Zhu[2], Mayte Suaréz-Fariñas[2,5], Steven M Wolinsky[4], Andrew Kasarskis[2] & Eva Harris[1,*]

## Abstract

Chikungunya virus (CHIKV) is a mosquito-borne alphavirus that causes global epidemics of debilitating disease worldwide. To gain functional insight into the host cellular genes required for virus infection, we performed whole-blood RNA-seq, 37-plex mass cytometry of peripheral blood mononuclear cells (PBMCs), and serum cytokine measurements of acute- and convalescent-phase samples obtained from 42 children naturally infected with CHIKV. Semi-supervised classification and clustering of single-cell events into 57 sub-communities of canonical leukocyte phenotypes revealed a monocyte-driven response to acute infection, with the greatest expansions in "intermediate" CD14++CD16+ monocytes and an activated subpopulation of CD14+ monocytes. Increases in acute-phase CHIKV envelope protein E2 expression were highest for monocytes and dendritic cells. Serum cytokine measurements confirmed significant acute-phase upregulation of monocyte chemoattractants. Distinct transcriptomic signatures were associated with infection timepoint, as well as convalescent-phase anti-CHIKV antibody titer, acute-phase viremia, and symptom severity. We present a multiscale network that summarizes all observed modulations across cellular and transcriptomic levels and their interactions with clinical outcomes, providing a uniquely global view of the biomolecular landscape of human CHIKV infection.

**Keywords** chikungunya; CyTOF; immune profiling; pediatric; RNA-seq
**Subject Categories** Genome-Scale & Integrative Biology; Microbiology, Virology & Host Pathogen Interaction; Molecular Biology of Disease
**Mol Syst Biol. (2018) 14: e7862**

## Introduction

Chikungunya (CHIKV) is a re-emerging mosquito-borne alphavirus that causes explosive epidemics throughout tropical regions of the world (Weaver & Lecuit, 2015). Chikungunya is mainly transmitted by *Aedes aegypti* and *Aedes albopictus* mosquitoes, the same vectors that transmit dengue (DENV) and Zika (ZIKV) viruses. Phylogenies of CHIKV indicate that urban endemic strains originated from several transmission events out of enzootic, sylvatic cycles between non-human primates and arboreal mosquitoes in eastern Africa (Volk *et al*, 2010). In 2004, a large outbreak of chikungunya spread rapidly from Africa through the Indian Ocean and Asia to Papua New Guinea and islands in the Pacific; subsequently, the first autochthonous transmission in the Americas was reported in 2013, leading to a dramatic continent-wide epidemic, including the USA, in 2014–2015 (Nasci, 2014). Millions of cases have now been reported in at least forty countries (Suhrbier *et al*, 2012).

Unlike other arboviral diseases, such as dengue, the majority of infections with CHIKV are symptomatic, with typical manifestations consisting of abrupt onset of high fever, a diffuse body rash, and joint pain and inflammation (Couderc & Lecuit, 2015; Weaver & Lecuit, 2015). Debilitating joint-related symptoms can persist for years, mimicking rheumatoid or psoriatic arthritis in up to 50% of afflicted populations—the namesake characteristic of the disease, as "chikungunya" is a word in the Kimakonde language describing a bent posture (Chopra *et al*, 2008; Miner *et al*, 2015; Weaver & Lecuit, 2015). Rarely, complications can occur that include encephalopathy, encephalitis, fulminant hepatitis, myocarditis, and multi-organ failure (Rolph *et al*, 2015). Mortality is rare and occurs in approximately 0.1% of cases (Rolph *et al*, 2015). Besides anti-inflammatories for symptomatic relief, there are no specific treatments available for chikungunya (Suhrbier *et al*, 2012; Weaver & Lecuit, 2015). Several vaccine candidates have reached preclinical

1   Division of Infectious Diseases and Vaccinology, School of Public Health, University of California Berkeley, Berkeley, CA, USA
2   Department of Genetics and Genomic Sciences, Icahn School of Medicine at Mount Sinai, New York, NY, USA
3   Human Immune Monitoring Center, Icahn School of Medicine at Mount Sinai, New York, NY, USA
4   Division of Infectious Diseases, Department of Medicine, Northwestern University Feinberg School of Medicine, Chicago, IL, USA
5   Department of Population Health and Science Policy, Icahn School of Medicine at Mount Sinai, New York, NY, USA
6   Laboratorio Nacional de Virología, Centro Nacional de Diagnóstico y Referencia, Ministerio de Salud, Managua, Nicaragua
    *Corresponding author. Tel: +1 510 642 4845; E-mail: eharris@berkeley.edu
    †These authors contributed equally to this work.

or phase I trials (Partidos *et al*, 2011; Weger-Lucarelli *et al*, 2014; Plante *et al*, 2015), but major commercial investment will be required to complete their development, and finding clinical sites to demonstrate efficacy is complex because of the unpredictable incidence and spread of the virus (Weaver & Lecuit, 2015). Until a vaccine or antiviral agent is available, prevention efforts remain focused on mosquito control (Weaver *et al*, 2012; Parashar & Cherian, 2014; Abdelnabi *et al*, 2015; Weaver & Lecuit, 2015).

Despite chikungunya's recent re-emergence in the Western Hemisphere, there are profound gaps in the understanding of CHIKV immunity and pathogenesis, including uncertainty over the roles of viral proteins and the myriad genetic and signaling factors involved in a successful or unsuccessful immune response (Sourisseau *et al*, 2007; Assunção-Miranda *et al*, 2013; Weaver & Lecuit, 2015), with particularly scarce information regarding pediatric cases (Teng *et al*, 2015). Chikungunya can infect many cell types, including skin fibroblasts, endothelial cells, primary epithelial cells, and human muscle satellite cells (Couderc & Lecuit, 2015; Lum & Ng, 2015). Reports of tropism in subsets of peripheral blood mononuclear cells (PBMCs) vary. Thus far, primary B and T cells have not been successfully infected *in vitro* (Sourisseau *et al*, 2007; Her *et al*, 2010; Teng *et al*, 2012). Although primary monocytes and macrophages only appear to be infectable at low efficiency *in vitro* (Sourisseau *et al*, 2007; Teng *et al*, 2012), CHIKV antigens have been detected in monocytes during acute infection (Her *et al*, 2010).

Monocytes and macrophages are thought to play a substantial role in the acute inflammatory response to CHIKV, as primate models show recruitment of these cell types, as well as natural killer (NK) cells, to infected tissues (Labadie *et al*, 2010). Monocytes can be further categorized into several subgroups based on differential expression of CD14 and CD16, including a $CD14^{+}CD16^{-}$ "inflammatory" subpopulation and a $CD14^{+}CD16^{+}$ subpopulation that has recently been differentiated into "intermediate" ($CD14^{++}CD16^{+}$) and "nonclassical" ($CD14^{+}CD16^{++}$) subtypes (Ziegler-Heitbrock *et al*, 2010). Murine studies show that the recruitment of "inflammatory" monocytes into infected tissues is CCR2-dependent; therefore, treatment of infected mice with bindarit (an inhibitor of CCR2 ligands CCL2, CCL7, and CCL8) resulted in reduced monocyte recruitment, joint swelling, and bone loss (Rulli *et al*, 2011; Chen *et al*, 2015). However, the role of some monocytes appears to be protective instead of inflammatory. For example, mice deficient for CCR2 (the receptor for CCL2) showed prolongation of arthritic disease corresponding with replacement of the monocyte/macrophage infiltrate in infected joints by neutrophils and eosinophils (Poo *et al*, 2014). In humans, however, details of the relationship between monocyte subpopulations, acute-phase pathogenesis, and chronic symptomatology remain poorly understood (Weaver & Lecuit, 2015; Burt *et al*, 2017).

The innate immune response, particularly via type I interferon (IFN) signaling, is important for control of CHIKV replication during the acute phase of infection (Schilte *et al*, 2010; Burt *et al*, 2017). CHIKV infection acutely induces high levels of IFN-α and IFN-γ release in both humans and model organisms (Labadie *et al*, 2010; Teng *et al*, 2015). In mouse models, type I IFNs control CHIKV replication by directly acting on non-hematopoietic cells, likely via activation of host sensors for viral RNA, such as RIG-I and MDA5 (Schilte *et al*, 2010). Additionally, either IRF3 or IRF7 signaling appears to be independently sufficient for

preventing lethality of CHIKV infection in adult mice (Schilte *et al*, 2012). In primary cell culture and mice, IFN-stimulated genes such as the OAS family and *RSAD2* (Viperin) appear to exert antiviral roles against CHIKV, although the details of these signal transduction pathways and their relative importance are unresolved (Burt *et al*, 2017).

Historically, the immune system has been described by evaluating individual components in isolation. This approach is often biased toward better-recognized phenotypes and pathways and is likely to miss globally significant patterns of interconnectivity, particularly across the multiple conjoint scales of the immune system, for example, transcriptional modulation within cells, resultant expansion and contraction of certain cell populations, and crosstalk between these immune cells and disparate tissues (Kidd *et al*, 2014). Genome-wide expression profiling using microarrays or RNA sequencing (RNA-seq), mass cytometry using cytometry time-of-flight (CyTOF), and cytokine profiling using a multiplex ELISA (Luminex) offer the capability to perform unbiased, systematic exploration of hundreds of thousands of changes transpiring within a particular perturbation of the immune system. Weighted coexpression and probabilistic causal network models can then synthesize data from "omics" assays into a map of quantitative relationships between all regulatory elements of a particular immune response, which is one of the goals of systems immunology (Germain *et al*, 2011; Arazi *et al*, 2013). Although biomolecular network models have demonstrated utility in finding causal gene modules and novel mechanisms for complex, inheritable human diseases (Chen *et al*, 2008; Emilsson *et al*, 2008; Zhang *et al*, 2013; Huan *et al*, 2015), because of the difficulty in acquiring data at the scale necessary for fitting these models, they remain relatively new in the field of infectious diseases. Nonetheless, network models have already helped map detailed regulatory circuits in hematopoiesis, transcriptional regulation of hundreds of leukocyte populations in mice, and viral sensing mechanisms in dendritic cells via Toll-like receptors (TLRs) (Kidd *et al*, 2014).

Previous observational studies of the immune response to CHIKV in natural human infections have typically concentrated on protein or gene expression levels of a small number of cytokines and inflammatory mediators (Ng *et al*, 2009; Chaaitanya *et al*, 2011; Chow *et al*, 2011; Kelvin *et al*, 2011; Teng *et al*, 2015), often producing conflicting results (Burt *et al*, 2017). Among studies that used cytometry, one group employed CyTOF to profile ten CHIKV patients, but their analysis focused almost entirely on T cells and included fewer markers for characterizing PBMCs (Miner *et al*, 2015). Our study, by contrast, employs a systems immunology approach, integrating three high-dimensional techniques to comprehensively profile the acute and convalescent phases of 42 pediatric cases of natural CHIKV infection from a hospital-based study in Managua, Nicaragua. We provide here a large RNA-seq study of CHIKV infection in humans, report CHIKV-induced modulations for nearly all PBMC subpopulations in humans, and, to our knowledge, present the first study that applies multiple profiling techniques to pediatric cases of CHIKV. We sampled whole blood from hospital patients with laboratory-confirmed CHIKV infection, comparing each patient's acute-phase sample (1–2 days post-symptom onset [p.s.o.]) against paired samples collected 2 weeks later (days 15–17 p.s.o.), after resolution of symptoms and viremia. We analyzed whole blood, PBMCs, and serum using RNA-seq, CyTOF, and

multiplex bead array ELISA (Luminex), respectively, employing a systematic, hypothesis-free approach for finding globally significant changes in cell subpopulation frequencies, gene expression, and serum cytokine/chemokine concentrations during acute CHIKV infection. We then searched for interactions within all of these data and with measurements of clinical outcomes, such as viral titer during the acute phase, severity of symptoms at presentation, and convalescent-phase anti-CHIKV antibody titer. Finally, to synthesize these interactions across the three scales examined, we present a multiscale network model that summarizes all correlations between gene modules, cell subpopulations, and clinical variables in this study.

# Results

### Clinical characteristics of study participants

Blood samples were collected as part of a hospital-based study of dengue and chikungunya in the Nicaraguan National Pediatric Reference Hospital in Managua, Nicaragua. Patients suspected of DENV or CHIKV infection were sampled and tested, and diagnosis of chikungunya was confirmed by real-time RT–PCR in the acute-phase sample. A total of 42 pediatric cases with detectable CHIKV viremia presenting to the hospital between November 2014 and October 2015 were included, from which acute (1–2 d p.s.o.) and convalescent (15–17 d p.s.o.) samples were obtained for analysis, for a total of 84 samples. The titer of anti-CHIKV antibodies was determined in acute- and convalescent-phase samples. In addition, we also collected blood samples at 3 ($n = 32$), 6 ($n = 26$), and 12 months ($n = 12$) p.s.o. to study potential associations with chronic CHIKV-induced arthritis. Although we lost several patients to follow-up at 3 ($n = 10$), 6 ($n = 16$), and 12 months ($n = 30$), none of the sampled patients progressed to chronic disease, and we therefore focused our study on the acute and convalescent (15–17 d p.s.o.) phases of CHIKV infection.

De-identified clinical data were obtained for all samples and are summarized in Table 1. Thirty-one (74%) of the patients were male, and 23 (57%) were between 9 and 14 years of age. Rash was the most common presenting symptom (41/42, 98%), followed by arthralgia (36/42, 86%). Most (16/42; 38%) of the patients were febrile (mean temperature, 38.3°C), and the average fever duration was 2.4 days. Of the cases studied, 36/42 (86%) resulted in hospitalization. We adapted a previous rubric for classifying chikungunya cases as less severe vs. more severe to determine the association between acute-phase severe manifestations and cytokine profile, gene expression, and cell population changes (Ng *et al*, 2009; Chow *et al*, 2011). In this study, a more severe case is defined by a peak temperature > 38.5°C or a nadir platelet count < 100,000 mm$^{-3}$. By this criterion, half of the pediatric cases (21/42, 50%) were considered more severe at the acute-phase sampling timepoint.

Whole-blood, serum, and PBMC samples were then analyzed for changes correlating with the acute and convalescent phases of infection, more severe and less severe cases, the convalescent anti-CHIKV antibody titer, and the acute-phase viral titer in serum (Fig 1). Finally, the resulting signatures and clusters were combined into a multiscale interaction network capturing the global landscape of immune responses to CHIKV.

**Table 1. Clinical characteristics of study population.**

| Phenotype/covariate | Participants, *N* = 42 |
|---|---|
| Gender | |
| Female, no. (%) | 11 (26) |
| Male, no. (%) | 31 (74) |
| Age | |
| Years, mean ± SD | 9.22 ± 4.5 |
| 1–4 years old, *n* (%) | 9 (21) |
| 5–8 years old, *n* (%) | 9 (21) |
| 9–14 years old, *n* (%) | 23 (57) |
| Signs or symptoms at enrollment | |
| Days post-symptom onset, mean ± SD | 1.41 ± 0.5 |
| Fever, mean temperature ± SD, °C | 38.3 ± 0.8 |
| Fever, mean duration ± SD, days | 2.4 ± 0.6 |
| High fever, peak temperature > 38.5°C, *n* (%) | 16 (38) |
| Retroorbital pain, *n* (%) | 7 (16) |
| Osteomuscular pain, *n* (%) | 26 (62) |
| Rash, *n* (%) | 41 (98) |
| Arthralgia, *n* (%) | 36 (86) |
| Myalgia, *n* (%) | 20 (48) |
| Headache, *n* (%) | 9 (21) |
| Abdominal pain, *n* (%) | 13 (31) |
| Vomiting, *n* (%) | 4 (10) |
| Fluid accumulation, *n* (%) | 14 (33) |
| Hospitalized, *n* (%) | 36 (86) |
| Laboratory values at enrollment | |
| Median platelet count, mm$^{-3}$ (range) | 199,000 (88,000–337,000) |
| Nadir platelet count < 100,000 mm$^{-3}$, *n* (%) | 8 (19) |
| Median white cell count, mm$^{-3}$ (range) | 8,140 (3,030–16,120) |
| Median monocyte % of WBCs (range) | 8.9 (4.1–14.6) |
| Median lymphocyte % of WBCs (range) | 39.7 (10.4–66.4) |
| Laboratory values at convalescent timepoint | |
| Days post-symptom onset, mean ± SD | 15.7 ± 0.6 |
| Median CHIKV Ab titer, dilutions (range) | 1,458 (232–7,794) |
| Severity categorization[a] | |
| More severe (%) | 21 (50) |
| Less severe (%) | 21 (50) |

WBC, white blood cell; CHIKV, chikungunya virus; Ab, antibody.
[a]Cases were categorized as more severe if the patient had either a peak fever of >38.5°C or a nadir platelet count of < 100,000 mm$^{-3}$.

## Acute infection associates with expansion of CD14$^+$CD16$^+$ monocytes

We used CyTOF to quantify 37 immune cell surface markers and the CHIKV envelope protein E2 in each of our samples. The high dimensionality of CyTOF data presents challenges for applying the traditional gating methods used in lower-dimensional flow cytometry.

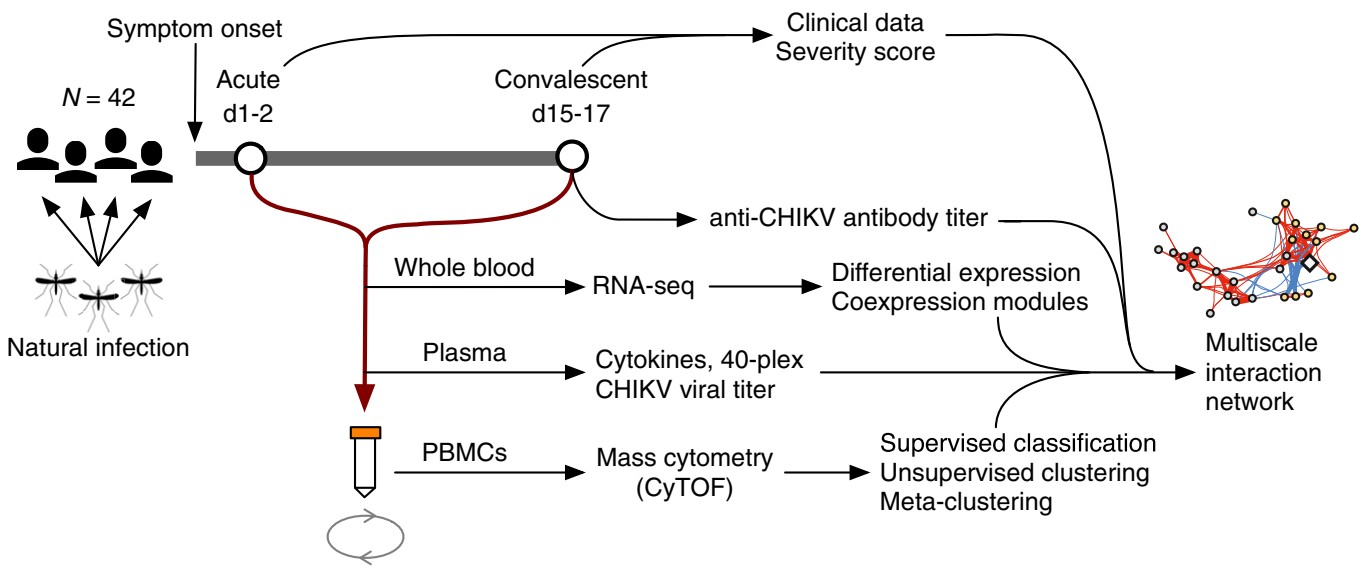

**Figure 1. Study design.**

Blood samples were obtained from 42 pediatric cases of natural chikungunya (CHIKV) infections at an acute (d1–2) and a convalescent (d15–17) timepoint, relative to reported symptom onset. Samples were separated into whole-blood, serum, and peripheral blood mononuclear cell (PBMC) aliquots for transcriptomic analysis, CHIKV viral titer assays, multiplex ELISA for cytokines, and mass cytometry (CyTOF). These data were combined with clinical data, including a severity score and a d15–17 anti-CHIKV antibody titer, to create a multiscale network of interactions during the observed course of CHIKV infection.

To address these challenges, we developed a sequential, semi-supervised approach to identify and classify immune cell populations in the CyTOF dataset. Manual gating and human-authored labels were first applied to a subset of the data to train a logistic regression classifier (called NodLabel) that was run on the remaining samples to broadly define nine major immune subsets in each of the patient samples. Figure 2A illustrates this process using a viSNE layout algorithm (Amir *et al*, 2013) for 2D visualizations of high-dimensional CyTOF data from a representative patient sample. To define additional heterogeneity within each of these broad subsets, we next applied Louvain/Phenograph (Levine *et al*, 2015) as an unsupervised clustering method to each NodLabel-identified subset in each patient sample. While the combination of the NodLabel classifier and Phenograph (which we term HybridLouvain) is a powerful approach to define phenotypic heterogeneity in a sample, a limitation of the approach is that the identified HybridLouvain communities are only applicable to a single patient sample. Therefore, we used a meta-clustering approach, termed MetaHybridLouvain and described further in Materials and Methods, to unify the community labels across multiple samples. We identified 26 communities of

canonical leukocyte populations across all acute- and convalescent-phase samples (Fig 2B). Hierarchical clustering by sample revealed two clusters that generally separated by timepoint (Jaccard index = 0.61, F-measure = 0.76), with no communities corresponding to any apparent contrasts in severity, convalescent anti-CHIKV antibody titer, age, sex, or acute-phase viral titer. Clustering by community frequency revealed a distinct contrast in $CD14^+CD16^+$ monocyte frequencies between the acute and convalescent phases (vertical axis, Fig 2B). This is the most expanded population at the acute timepoint (Fig 2C), and the difference was highly significant (q-value, aka false discovery rate-adjusted $P$ [FDR $P$] = $1.7 \times 10^{-18}$; moderated paired $t$-test, mixed-effects model). Other populations also comparatively upregulated during the acute phase were plasmacytoid dendritic cells (pDCs) (FDR $P = 1.0 \times 10^{-21}$) and $CD14^+$ monocytes (FDR $P = 5.4 \times 10^{-8}$), with five additional populations identified at a threshold false discovery rate (FDR) < 0.05 (Fig 2C). Ten populations were comparatively downregulated at the acute phase and thereby associated with the convalescent phase at FDR < 0.05 (Fig 2C), with the strongest difference being observed in CD1c dendritic cells (DCs) (FDR $P = 2.1 \times 10^{-26}$).

**Figure 2. CyTOF reveals signatures for acute CHIKV infection based on canonical immune cell phenotype clustering.**

A   Overview of the NodLabel procedure, using a viSNE layout of CyTOF single-cell events. Left side, point color indicates channel values for four example channels. Right side, traditional hierarchical gating was used on a subset of samples to identify 8 major immune compartments, which was then used to train a logistic regression classifier (Nod) that applied labels for canonical leukocyte phenotypes to all samples (NodLabel).

B   Heatmap of $\log_{10}$-scaled PBMC community frequencies for all samples (n = 42 patients × 2 timepoints = 84 samples). Clinical variables are depicted for all samples across the top of the heatmap; convalescent post-symptom onset anti-CHIKV antibody titer and viral titer (which was measured during the acute phase) are both in units of $\log_{10}$ dilutions. Hierarchical clustering (using complete linkage) was applied to both samples (X-axis) and communities (Y-axis). Four major clusters of communities and two major clusters of samples (largely separating acute and convalescent samples) are highlighted.

C   Fold changes in $\log_{10}$ frequencies for PBMC communities between acute- and convalescent-phase samples, filtered to a significance threshold of FDR < 0.05 (moderated paired $t$-test, mixed-effects model). Error bars represent the 95% confidence interval. The acute-phase frequency for each community is depicted with the purple heatmap. *FDR-adjusted $P$ (FDR $P$) < 0.05; ***FDR $P$ < 0.001.

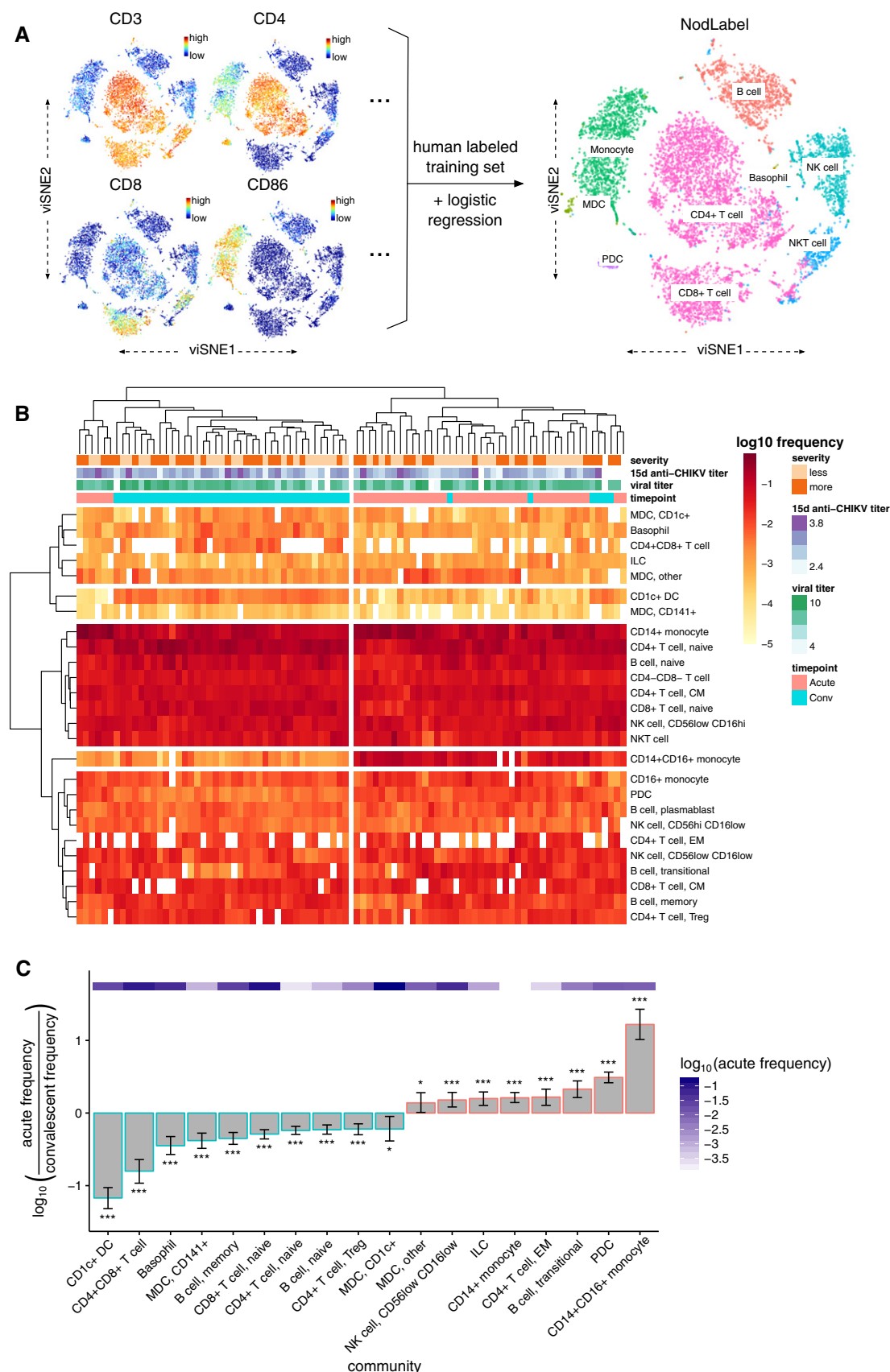

**Figure 2.**

**Monocytes, dendritic cells, and B cells are positive for CHIKV envelope protein E2 during acute infection**

Classifying CyTOF events into only the canonical leukocyte populations ignores much of the richness of these data, which can reveal previously unrecognized diversity and heterogeneity within each of these populations. An advantage of our MetaHybridLouvain approach (Fig 3A) when compared to most other automated classifiers (Aghaeepour *et al*, 2013; Lee *et al*, 2017) is that it allows for unbiased identification of phenotypically heterogenous communities within each of the canonical immune subsets (sub-communities) (Samusik *et al*, 2016); e.g., a *sub-community* of $CD14^+$ monocytes is defined by a specific, reproducible marker expression pattern for a subset of the population across multiple samples. We identified up to nine sub-communities within each of the pre-defined canonical immune populations (Fig 3B and C), producing a total of 57 sub-communities. More detailed results of MetaHybridLouvain for the representative sample used in Figs 2A and 3B are depicted in Appendix Fig S1.

Having dissected the cellular heterogeneity of the samples at high resolution, we went on to identify leukocyte populations that display comparatively high levels of CHIKV envelope protein (E2) on their surface during the acute phase of infection. Qualitatively, in the representative patient sample, CHIKV E2 protein was expressed on distinct populations of leukocytes as depicted in the viSNE layout (Fig 3D)—in particular monocytes, dendritic cells (DCs), and B cells (compare with Fig 2A). Quantitatively, across all samples, monocytes and DCs displayed the strongest contrasts in mean CHIKV E2 channel expression values per sample between acute and convalescent timepoints (Fig 3E). CHIKV E2-positive cell populations largely fell along canonical leukocyte phenotype boundaries, with all three sub-communities of $CD14^+$ monocytes (FDR $P = 1.3 \times 10^{-11}$, $6.5 \times 10^{-12}$, $1.4 \times 10^{-10}$, moderated paired *t*-test, mixed-effects model), both sub-communities of CD1c mDCs (FDR $P = 8.5 \times 10^{-5}$ and $1.2 \times 10^{-8}$), all CD1c DCs (FDR $P = 1.4 \times 10^{-7}$), and both sub-communities of $CD14^+CD16^+$ monocytes (FDR $P = 2.4 \times 10^{-6}$ and $3.2 \times 10^{-8}$) identified as significantly more CHIKV-positive during

the acute phase. Interestingly, although the differences were less pronounced, two sub-communities of B cells were also observed to express significantly higher CHIKV E2 protein during acute infection: the only community of memory B cells (FDR $P = 1.4 \times 10^{-8}$) and the fourth of the four sub-communities of naïve B cells (FDR $P = 8.7 \times 10^{-6}$). This sub-community is characterized by a higher expression of CXCR5 as compared to the other three sub-communities of naïve B cells (Appendix Fig S2); however, the expression of other markers is similar to the first sub-community of naïve B cells, albeit a much lower expression of CCR4, CXCR3, and CD80. Although CHIKV E2 protein expression only correlates with (and does not establish) tropism of the virus, our data suggest that among PBMCs, CHIKV preferentially infects monocytes and DCs, while displaying lower but substantial affinity for B cells.

**$CD14^+$ and $CD14^+CD16^+$ monocyte sub-communities exhibit contrasting behaviors during acute infection**

Although $CD14^+CD16^+$ monocytes expand during the acute phase of infection, community subclustering provides more details on particular sub-communities that associate with the acute phase. Hierarchical clustering of samples by sub-community frequencies separates the samples by timepoint more effectively (Jaccard index = 0.87, F-measure = 0.93) than canonical population frequencies alone (Fig 3F). There was no apparent clustering of samples that corresponded to contrasts in clinical severity, convalescent anti-CHIKV antibody titer, or acute-phase viral titer. Stratifying by the acute and convalescent phases, there were no significant differences in any sub-community frequencies between more severe and less severe cases at FDR < 0.1. Within either timepoint, there were also no significant correlations between sub-community frequencies and log-transformed acute-phase viral titers at FDR < 0.1. There was, however, a single significant correlation between $CD14^+CD16^+$ monocyte sub-community 1 at the acute phase and the convalescent anti-CHIKV antibody titer (FDR $P = 0.0050$, Spearman's $\rho = 0.60$; see Appendix Fig S3A). There were significant correlations between the convalescent anti-CHIKV

---

**Figure 3.    Monocytes, dendritic cells, and B cells express CHIKV proteins during acute CHIKV infection, but only specific monocyte sub-communities undergo expansion.**

A    Overview of the MetaHybridLouvain procedure.

B    viSNE layout of CyTOF single-cell events from the same representative sample as Fig 2A, but colored according to the 26 canonical leukocyte phenotypes detected by MetaHybridLouvain. Unique sub-communities of each canonical phenotype are differentiated by the numerals used as point marks. The categorizations from NodLabel (as in Fig 2A) are provided as text labels. Note that over-plotting has been disallowed, so this is a sampling of the single-cell events.

C    Number of sub-communities detected by MetaHybridLouvain for each of the canonical leukocyte phenotypes.

D    viSNE layout of CyTOF single-cell events from the same representative sample as Figs 2A and 3A, but with points colored according to the CHIKV channel. By qualitative comparison with Fig 2A, monocytes, myeloid dendritic cells (mDCs), and B cells (labeled) have the highest CHIKV envelope protein (E2) expression on the cell surface.

E    Differences in CHIKV E2 protein expression between acute-phase and convalescent-phase samples per MetaHybridLouvain sub-community. Sub-communities are ordered by largest to smallest difference and filtered to sub-communities where the median of the channel means per sample was higher in the acute-phase samples at a significance threshold of FDR < 0.05. Sub-communities are named by their parent canonical community name plus an arbitrary number, up to the numbers given in (C). Lower and upper hinges of the boxplot correspond to the first and third quartiles; whiskers extend to the most distant values no further than 1.5× the interquartile range from the hinge.

F    Heatmap of $\log_{10}$-scaled PBMC sub-community frequencies for all samples ($n = 42$ patients × 2 timepoints = 84 samples). Clinical variables are depicted for all samples across the top of the heatmap; convalescent post-symptom onset anti-CHIKV antibody titer and viral titer (which was measured during the acute phase) are both in units of $\log_{10}$ dilutions. Hierarchical clustering (using complete linkage) was applied to both samples (*X*-axis) and sub-communities (*Y*-axis). Four major clusters of sub-communities and two major clusters of samples (largely separating acute and convalescent samples) are highlighted.

G    Fold changes in $\log_{10}$ frequencies for PBMC sub-communities contrasted between acute- and convalescent-phase samples, filtered to a significance threshold of FDR < 0.05 (moderated paired *t*-test, mixed-effects model). Error bars represent the 95% confidence interval. The acute-phase frequency for each sub-community is depicted with the purple heatmap. *FDR-adjusted *P* (FDR *P*) < 0.05; **FDR *P* < 0.01, ***FDR *P* < 0.001.

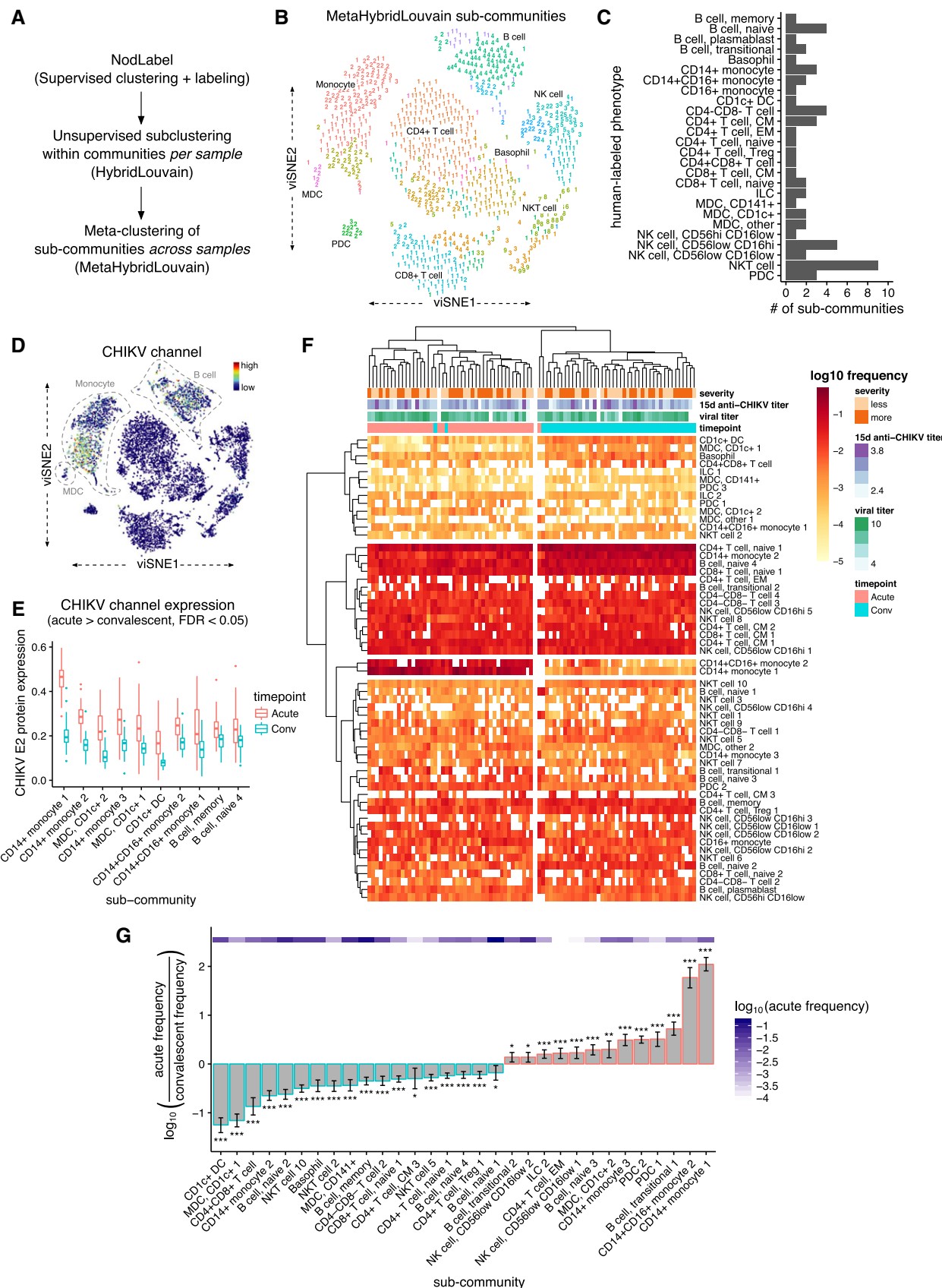

**Figure 3.**

antibody titer and two sub-communities: again, $CD14^+CD16^+$ monocyte sub-community 1 (FDR $P = 0.035$, $\rho = 0.51$) and central memory $CD4^+$ T cell sub-community 2 (FDR $P = 0.035$, $\rho = -0.52$; see Appendix Fig S3B).

Among sub-communities significantly expanded during the acute phase, two particular expansions separated from the others by an order of magnitude (Fig 3G), specifically sub-community 1 of $CD14^+$ monocytes (FDR $P = 2.6 \times 10^{-45}$, moderated paired *t*-test, mixed-effects model) and sub-community 2 of $CD14^+CD16^+$ monocytes (FDR $P = 7.8 \times 10^{-26}$). When examining the other two sub-communities of $CD14^+$ monocytes, one was also expanded during the acute phase but to a lesser extent (sub-community 3, FDR $P = 4.2 \times 10^{-12}$), while the other instead was expanded during the convalescent phase (sub-community 2, FDR $P = 1.9 \times 10^{-21}$). At FDR $< 0.1$, sub-community 1 of $CD14^+CD16^+$ monocytes, which is the only other sub-community of $CD14^+CD16^+$ monocytes, was not significantly different across timepoints. Other sub-communities associating with the convalescent phase at FDR $< 0.05$ included mDCs, CD1c DCs, B cells, T cells, and basophils (Fig 3G).

Since different sub-communities of $CD14^+CD16^+$ and $CD14^+$ monocytes displayed distinctive responses to acute CHIKV infection, we looked for marker differences that could better define these sub-communities. Examination of the two $CD14^+CD16^+$ monocyte sub-communities (Fig 4A) revealed that among all significant marker differences, sub-community 1 had higher CD16 expression (FDR $P = 6.7 \times 10^{-40}$, moderated paired *t*-test, mixed-effects model) and sub-community 2 had higher CD14 expression (FDR $P = 6.6 \times 10^{-11}$). This corresponded to sub-communities commonly called "nonclassical" $CD14^+CD16^{++}$ and "intermediate" $CD14^{++}CD16^+$ monocytes (Ziegler-Heitbrock *et al*, 2010; Wong *et al*, 2011), implying that in our study, "intermediate" monocytes were selectively and substantially expanded during acute CHIKV infection while

"nonclassical" monocytes were unchanged. Significant contrasts in the expression of many other surface markers at FDR $< 0.05$ (Appendix Fig S4) and the consistent identification of these patterns across the majority of samples (Appendix Figs S5 and S6) confirmed the distinction between these sub-communities.

Among the three sub-communities of $CD14^+$ monocytes, we discovered two that were associated with acute infection, including one with a previously unreported phenotype (sub-community 3). Sub-community 1 (the sub-community most strongly associated with acute infection and also expressing the highest levels of CHIKV E2 protein) was characterized by having relatively higher levels of CD123, $CX_3CR1$, CD86, and CD54 expression (Fig 4B), generally consistent with a more activated phenotype relative to sub-community 2, which was more prevalent during convalescence. In monocyte sub-community 2, CCR7 and CD40 were particularly downregulated compared to monocyte sub-communities 1 and 3. Monocyte sub-community 3 was also expanded during acute infection, though at a much lower frequency than monocyte sub-community 1, and displayed similar levels of CD40 and CCR7, consistent with an activated phenotype. Interestingly, however, this subset also exhibited comparatively high expression of markers that are not classically associated with monocytes, particularly the chemokine receptor CCR4, as well as CXCR3 and CCR6 (Fig 4B). We confirmed that this sub-community did not express canonical markers associated with other major cell types (such as T cells or B cells) to verify that it did not represent an artifact of cell–cell doublets, and although it is a rare subtype—approximately 1% of all monocytes—its presence was further confirmed via manual re-gating (Appendix Fig S7). Significant contrasts in the expression of many surface markers at FDR $< 0.05$ (Appendix Fig S8) and a consistent pattern for the phenotype identified across the majority of samples (Appendix Figs S9–S11) verified the distinction between

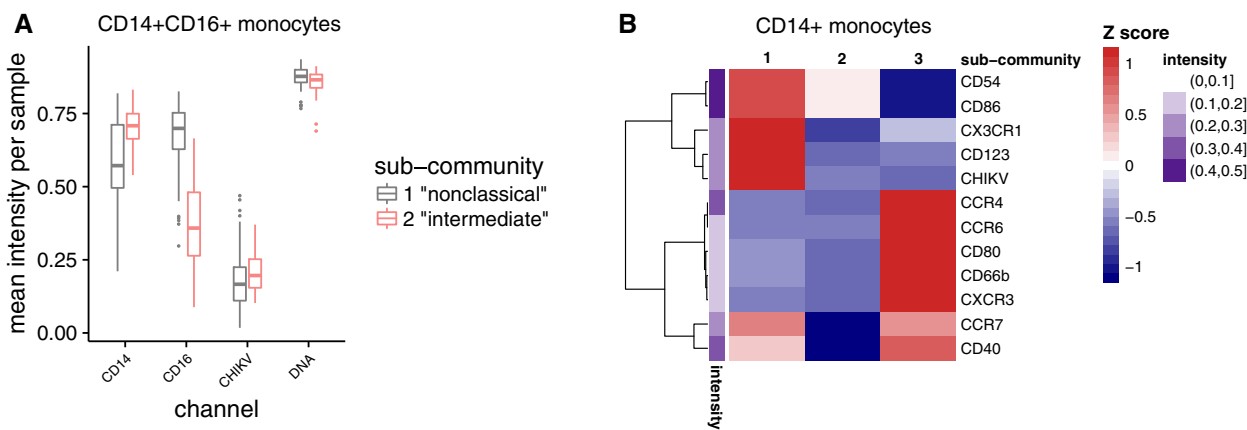

**Figure 4. Marker expression differences between sub-communities of $CD14^+CD16^+$ monocytes and $CD14^+$ monocytes.**

A  Relative expression of $CD14^+$ and $CD16^+$ in $CD14^+CD16^+$ sub-communities, depicted as boxplots of the mean expression levels for all samples ($n = 42$ patients $\times$ 2 timepoints = 84 samples), indicates that sub-community 1 has a $CD14^+CD16^{++}$ (aka "nonclassical") phenotype, while sub-community 2 has a $CD14^{++}CD16^+$ (aka "intermediate") phenotype. Differences shown here are significant at FDR $< 0.05$ (moderated paired *t*-test, mixed-effects model); for a view of all differences significant at this threshold, see Appendix Fig S3. Box hinges and whiskers are calculated as in Fig 3E.

B  Heatmap of relative expression of markers that most differentiate the sub-communities of $CD14^+$ monocytes by difference in mean expression levels for all samples ($n = 42$ patients $\times$ 2 timepoints = 84 samples). Markers are filtered to those different at a significance threshold of FDR $< 0.05$ (moderated paired *t*-test, mixed-effects model) and fold change $> 1.3$; for a view of all differences significant at FDR $< 0.05$, see Appendix Fig S8. The mean intensity of each channel across all sub-communities is shown by the adjacent purple heatmap.

these sub-communities. Differential expression of genes for these surface markers across the two timepoints was confirmed by RNA-seq (Table EV1; complete analysis presented later in this section). Given the strongly contrasting associations of these sub-communities with the phase of infection, these data suggest that unappreciated heterogeneity within CD14$^+$ monocyte phenotypes may enable different roles for sub-communities of these monocytes during CHIKV infection.

**Monocyte-associated cytokine concentrations increase during acute infection**

To profile the effect of CHIKV on circulatory markers for inflammation and immune signaling, we used a multiplex ELISA-based assay (Luminex) to measure serum concentrations of 40 cytokines, chemokines, and growth factors in all 84 samples. In our study, after adjusting for plate-specific batch effects, twelve cytokines showed highly significant differences (using Benjamini–Hochberg adjustment; Wilcoxon signed-rank test) across acute and convalescent timepoints at FDR $P < 0.001$ (Fig 5A–F). The strongest contrast was for the monocyte chemoattractant CCL2 (FDR $P = 3.6 \times 10^{-11}$).

Significant increases were also observed for IL-10 (a monocyte-secreted anti-inflammatory cytokine), CXCL10 (an IFN-γ-inducible monocyte-secreted cytokine), IFN-γ, IFN-α2, TNF-α, CXCL8 (IL-8), IL-1α, IL-15, G-CSF, GM-CSF, CCL4, CCL5, CCL11, CX₃CL1, IL-6, IL-1RA, IL-12p40, and IL-2. There was only one significant difference observed among growth factors, namely TGF-α (FDR $P = 0.026$; Appendix Fig S12). In addition, there was only one analyte concentration found to be significantly lower during acute infection, namely IL-17A (FDR $P = 0.0085$; Fig 5E).

To determine whether cytokine levels could be associated with changes in specific cell sub-communities, we correlated log-scaled Luminex analyte concentrations with log-scaled sub-community frequencies (using Spearman's ρ). Hierarchical clustering revealed that acute vs. convalescent shifts in cytokine and growth factor concentrations tended to correlate with each other rather than with any of the shifts in CyTOF-identified subpopulation frequencies (Appendix Fig S13). This remained unchanged when stratifying into acute-phase (Appendix Fig S14) or convalescent-phase (Appendix Fig S15) values alone. Since the most pronounced expansions during the acute phase involved monocyte subpopulations, we then performed a more focused analysis of Spearman's correlations

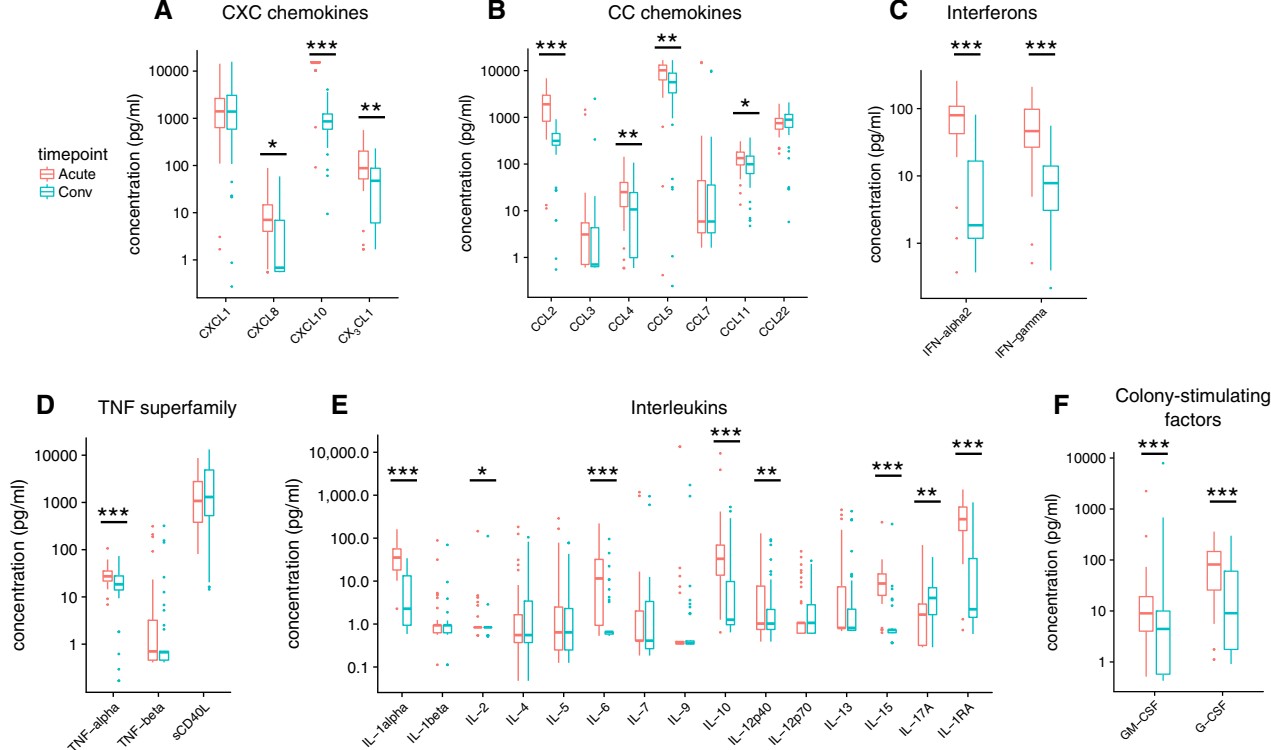

**Figure 5.  Differences in serum cytokine and chemokine levels between acute and convalescent phases of CHIKV infection.**

A  CXC chemokines.
B  CC chemokines.
C  Interferons.
D  TNF superfamily cytokines.
E  Interleukins.
F  Colony-stimulating factors.

Data information: $n = 42$ pairs of samples, 1 pair per patient. Wilcoxon signed-rank test was used to determine statistical significance, with a Benjamini–Hochberg adjustment for FDR (40 comparisons). *$P < 0.05$, **$P < 0.01$, ***$P < 0.001$. Growth factors are shown in Appendix Fig S12. Box hinges and whiskers are calculated as in Fig 3E.

between all cytokines and monocyte subpopulations, stratifying by timepoint to capture potential regulatory relationships rather than the primary contrast of the study. Within acute-phase samples, cytokines generally had varied correlations with each of the monocyte subpopulations, but notably, the monocyte chemoattractant CCL2 positively correlated with all monocyte subpopulations across both timepoints, and furthermore, its correlations with monocyte subpopulations significantly differed from the correlations against all other cytokines in convalescent-phase samples (Appendix Fig S16; FDR $P = 0.015$, Mann–Whitney $U$). These consistently positive correlations and the strength of its upregulation suggest a relatively important regulatory role for CCL2 on monocyte populations during the progression of CHIKV infection.

## Acute infection associates with upregulated transcription of monocyte-associated cytokine genes

We used RNA-seq to measure global transcriptional changes during CHIKV infection. Since previous studies of gene and protein changes during acute CHIKV infection targeted cytokines, chemokines, and innate immunity mechanisms (Ng et al, 2009; Hoarau et al, 2010; Chow et al, 2011; Wauquier et al, 2011; Teng et al, 2015), we first present results for differentially expressed genes in these pathways for comparison with our Luminex data and the literature, before moving to a global analysis.

Table EV2 shows that across two timepoints, log-scaled serum protein concentrations for the significantly CHIKV-upregulated cytokines (as measured by Luminex; Fig 5) did not consistently correlate per sample with whole-blood gene expression for corresponding genes, using voom-normalized expression levels. Since this could be due to different "baseline" serum cytokine or cytokine gene expression levels, we repeated the analysis with all acute-phase measurements normalized against the convalescent-phase measurements, but the result did not change (Table EV2, 3rd column). This suggests that the regulation of serum cytokine levels is not primarily driven by transcriptional changes in leukocytes, but could involve substantial expression from other tissues and secretory and protein-level regulatory processes.

Among CXC- and CC-chemokine subfamilies, we observed transcriptional upregulation of CXCL10, CXCL11, CCL2, and CCL8 (Appendix Fig S17). Of these, monocyte-secreted CXCL10 and monocyte chemoattractant CCL2 concur with the changes in serum cytokine concentrations described above (Fig 5A and B), and CCL8, aka MCP2 (whose gene product was not measured with Luminex), is notable for being another monocyte chemoattractant. Interestingly, although serum IFN-α and IFN-γ levels were significantly elevated during acute infection (Fig 5C), none of the interferon family genes were differentially expressed at these thresholds (Appendix Fig S17). Concordant with the significant increase in serum TNF-α concentration (Fig 5D), significantly upregulated TNF superfamily genes included TNFSF15, TNFSF10, and TNFSF13B (Appendix Fig S17). While upregulation of IL10 gene expression was concordant with the serum cytokine measurements, in contrast to those data, we did not observe differential expression of CXCL8 (aka IL8; Appendix Fig S17 and Fig 5E). We also examined differential expression of genes during acute infection for known components of innate immunity pathways annotated in KEGG (Ogata et al, 1999) (Appendix Figs S18–S23). Both RIG-I and MDA5, which are

cellular sensors for viral RNA, were significantly upregulated during acute infection (Appendix Fig S18). Within the JAK/STAT signaling pathway, JAK/STAT genes were significantly transcriptionally upregulated (Appendix Figs S19 and S20). Among Toll-like receptor (TLR) genes, TLR1, TLR2, TLR3, TLR5, and TLR7/8 were all significantly upregulated (Appendix Fig S21).

Of the TNF superfamily receptors, TNFR1 pathway intermediates were generally more upregulated during acute infection than TNFR2 pathway intermediates (Appendix Fig S22). Of the interferon regulatory factor (IRF) genes annotated in KEGG, expression of IRF7 was significantly upregulated, and interestingly, all downstream transcriptional targets of IRF9 annotated in KEGG (MX1, OAS genes, ADAR, and PML) were consistently upregulated during acute infection (Appendix Fig S23). (Fold-change values for all quantified genes and FDR P-values are provided in Table EV3.) In general, our observed modulations of human innate immunity pathways were comparable to gene-level transcriptomic effects reported for a mouse model in a recent study (Wilson et al, 2017) (see Discussion).

## Concordance between cell composition changes estimated from gene expression data and CyTOF sub-communities

Next, we wanted to compare the cell composition changes as determined by CyTOF with possibly downstream changes in whole-blood gene expression in the RNA-seq data. For this analysis, we derived a list of genes that are upregulated or downregulated in the acute phase of CHIKV infection (adjusted P-value < 0.001, modified paired t-test; Table EV3) compared to the convalescent phase. We then compared the expression levels of these genes across a panel of hematopoietic cells of different lineages (Novershtern et al, 2011). We observed that genes with higher expression in the acute phase tend to be overexpressed in granulocytes and monocytes while genes with higher expression in the convalescent phase tend to be overexpressed in T cells and B cells (Appendix Fig S24). As shown in Appendix Fig S25, computational estimation of cell components derived from the gene expression data using two independent methods (Abbas et al, 2005; Newman et al, 2015) recapitulates the cell sub-community shifts seen in our CyTOF data: correlations between the same or similar cell types across the two data types are generally much higher (red regions) than between different cell types (blue regions). This demonstrates an overall consistency between gene expression and CyTOF data in this study when used independently to estimate cell subpopulation frequencies and suggests that the main contribution to differential gene expression in whole blood was indeed the change in underlying leukocyte composition.

## Transcriptomic signatures for acute infection, severity, viral titer, and immunogenicity

RNA-seq enables the estimation of transcript abundances and transcript-level differential expression analyses that may offer insights not available from gene-level quantification (Anders et al, 2012; Trapnell et al, 2013). We obtained a strong transcriptional signature for timepoint (acute vs. convalescent transcript abundances), with 8,008 transcripts differentially expressed at FDR < 0.05 and FCH > 2 (Fig 6A). The top differentially expressed transcripts (DETs), ordered by P-value, were products of the EIF4B, XAF1, HERC6,

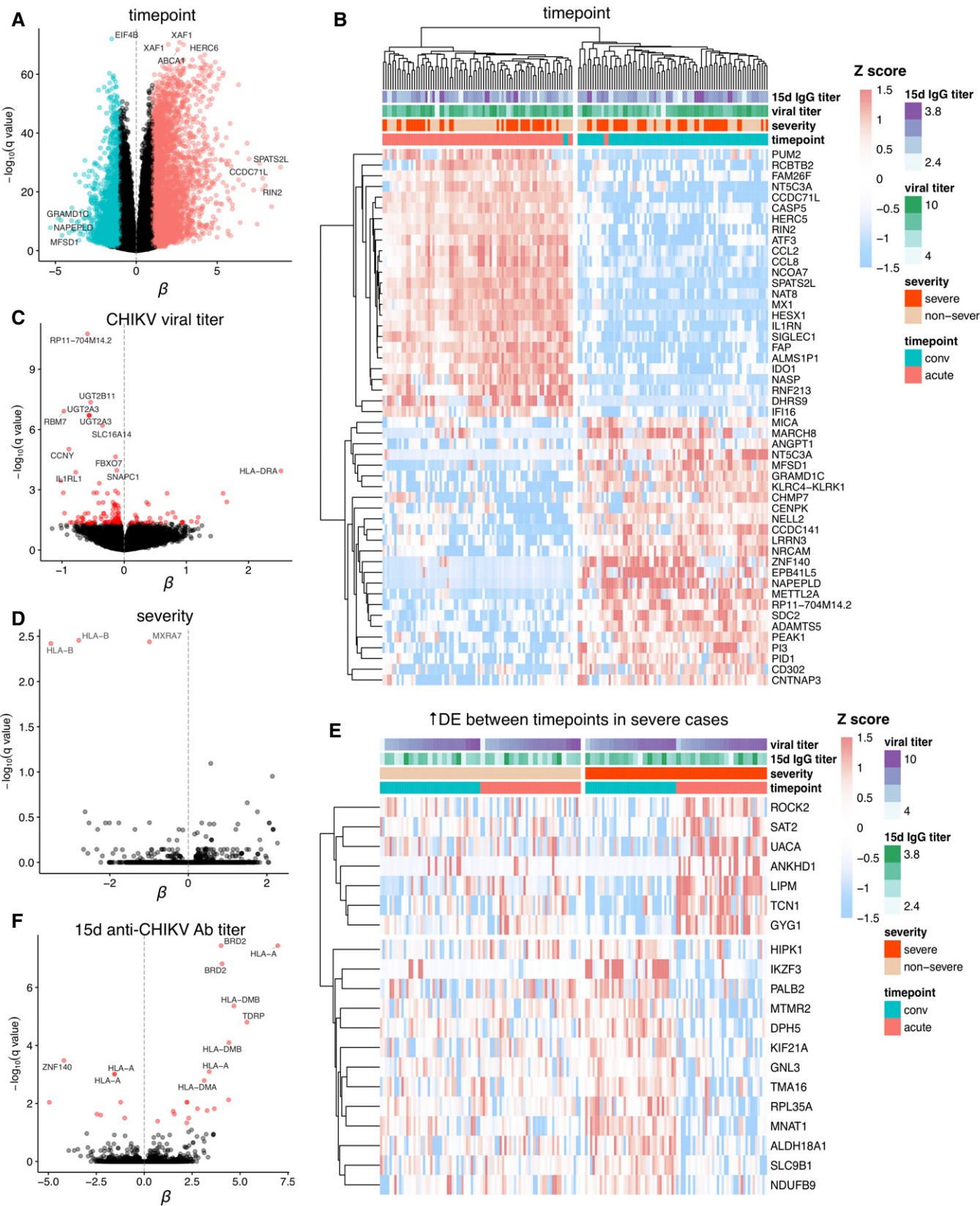

**Figure 6.**

**Figure 6.  Differentially expressed host transcripts across timepoint and clinical variable contrasts in natural CHIKV infections.**

Human whole-blood transcriptomic signatures for timepoint, CHIKV viral titer at the acute phase, symptom severity, and convalescent (15 d post-symptom onset) anti-CHIKV antibody (Ab) titer across the two observed phases of natural CHIKV infection. Expression was measured with RNA-seq and quantified at the transcript level, with all models of differential expression adjusting for patient age and gender as covariates. For all panels, 42 patients were sampled at 2 timepoints, each with 2 technical replicates; *q*-values are Benjamini–Hochberg-adjusted *P*-values from a moderated paired *t*-test under the mixed-effects model.

A   Volcano plot of differentially expressed host transcripts between acute- and convalescent-phase samples, with negative $\log_{10}$-scaled *q*-values on the *Y*-axis, and the fitted *β* coefficient for each transcript (corresponding to modeled $\log_2$ fold change) on the *X*-axis. Transcripts to the right of the vertical dashed line were comparatively upregulated in acute-phase samples, while transcripts to the left were upregulated during the convalescent phase. Transcripts that pass FDR < 0.05 and fold change > 2 are colored salmon and turquoise for acute-associated and convalescent-associated transcripts, respectively. Top transcripts by *q*-value or fold change are individually labeled by their corresponding gene symbol.
B   Heatmap of expression in units of Z-scores per transcript for the top 50 differentially expressed host transcripts between acute- and convalescent-phase samples. Clinical variables are depicted for all samples across the top of the heatmap; convalescent post-symptom onset anti-CHIKV antibody titer and viral titer (which was measured during the acute phase) are both in units of $\log_{10}$ dilutions. Hierarchical clustering (using complete linkage) was applied to both samples (*X*-axis) and transcripts (*Y*-axis). Two major clusters of samples (largely separating acute and convalescent samples) are highlighted.
C   Volcano plot as in (A) but for differentially expressed host transcripts between samples with higher and lower CHIKV viral titer. Transcripts to the right of the vertical dashed line are associated with higher viral titer, while transcripts to the left are associated with lower viral titer. Transcripts significant at FDR < 0.05 are colored red.
D   Volcano plot as in (C) but for differentially expressed host transcripts between patients with more severe and less severe acute-phase symptoms. Transcripts to the right of the vertical dashed line were comparatively upregulated in more severe cases, while transcripts to the left were upregulated in less severe cases.
E   Heatmap of expression as in (B) but for 20 differentially expressed host transcripts (all significant at FDR < 0.05) under a model that includes a severity–timepoint interaction. Transcripts have been filtered to those differentially expressed between the timepoints in more severe cases but not in less severe cases.
F   Volcano plot as in (C) but for differentially expressed host transcripts between patients with higher and lower convalescent (15 d post-symptom onset) anti-CHIKV antibody (Ab) titers. Transcripts to the right of the vertical dashed line were comparatively upregulated in patients with a higher convalescent anti-CHIKV antibody titer, while transcripts to the left were upregulated in patients with a lower convalescent anti-CHIKV antibody titer.

*ABCA1*, *IFI44L*, and *IFI44* genes. Hierarchical clustering of the samples by quantification of the top 50 significant DETs (by greatest FCH) readily separated samples by timepoint, with only 4/160 (2.5%) of samples misclassified between the two major clusters (Fig 6B). Adding the log-scaled acute-phase CHIKV viral titer as a variable to the model of transcript expression produced a separate and substantial signature for transcription correlating with viral titer (Fig 6C), with 177 DETs at FDR < 0.05. In Fig 6C, an increase in transcription corresponding to higher viral titers were modeled as a positive fixed-effect coefficient *β*. Top DETs for viral titer (by *P*-value) were products of the *RP11-704M14.2*, *UGT2B11*, *RBM7*, *UGT2A3*, and *SLC16A14* genes, with all of these genes downregulated in cases with higher viremia.

We assessed the composition of these two large signatures with Enrichr (Kuleshov *et al*, 2016), which performs standard gene set enrichment analyses for many functional annotation libraries. The 4,163 unique genes in the aforementioned timepoint DET signature were most significantly enriched for gene ontology (GO), PANTHER, and Reactome annotations related to viral transcription, apoptosis signaling, innate immunity signaling, and translation (Table 2). We found 681 unique genes in the viral titer DET signature (re-thresholded at FDR < 0.1) that were most significantly enriched for terms related to leukocyte activation, T cell activation, and both the innate and adaptive immune systems (Table 2). These enrichments suggest that the timepoint of infection sensibly corresponds uniquely with a contrast in genes related to viral transcription and translation, while the level of viremia instead uniquely corresponds with activation of the adaptive immune system.

Although the phase of infection and the level of viremia were expected to produce strong transcriptional signatures, we sought potential signatures for downstream clinical outcomes, such as the severity of acute-phase symptoms or the convalescent anti-CHIKV antibody titer, a correlate for humoral immunogenicity. For symptom severity, adding the more severe vs. less severe categorization of cases to the model produced a very small differential expression

signature of 3 transcripts at FDR < 0.05 and FCH > 1.5 (Fig 6D), with *P*-values for the top three transcripts diverging from the expected null distribution (Appendix Fig S26). Two of these transcripts were from *HLA-B*, one of which is its canonical protein-coding transcript, and the other of which is a retained intron; the third is the canonical transcript of *MXRA7*, which encodes a poorly characterized single-pass membrane protein. By design, this model only reveals DETs that correlate with severity across *both* timepoints of infection; therefore, we added a timepoint–severity interaction to the model and discovered 43 transcripts with a significant interaction term (FDR < 0.05, FCH > 1.5).

Figure 6E shows a heatmap that focuses on 20 DETs between timepoints only in the more severe cases, i.e., transcripts differentially expressed during acute illness that simultaneously and specifically correlated with worse symptomatology. We found a similarly sized signature for the convalescent anti-CHIKV antibody titer, with 27 DETs at FDR < 0.05 (Fig 6F). Top-ranked transcripts by *P*-value were again notable for including HLA genes, with several transcripts of *HLA-A*, *HLA-DMB*, and *HLA-DMA* among the top ten transcripts. Fold-change values for all quantified transcripts for the timepoint, acute viral titer, symptom severity, and convalescent anti-CHIKV antibody titer contrasts and corresponding FDR *P*-values are provided in Tables EV4–EV7.

### Multiscale network analysis

To relate CHIKV-associated transcriptomic changes to changes in cell sub-community frequencies, serum cytokine concentrations, and clinical variables, we identified coexpression patterns among sets of genes to create coexpression network modules (coEMs) using whole-genome coexpression network analysis (WGCNA) (Zhang & Horvath, 2005). The coEMs could then be correlated with other variables to capture the genomic co-regulatory structure from biological variability present across and within the timepoints. We identified 92 coEMs, which were named after arbitrary colors (Fig 7A and B).

**Table 2.   Gene set enrichment analysis of DET signatures.**

| Gene set[a](threshold) | Annotation set | Term | Overlap | q-value[b] | Z-score | Combined score[c] |
|---|---|---|---|---|---|---|
| 4163 DETs for timepoint: acute vs. convalescent (FCH > 2, q < 0.05) | GO Biological Process 2015 | Viral transcription | 71/84 | < 0.001 | −2.11 | 152 |
| | | Translational termination | 71/89 | < 0.001 | −2.11 | 138 |
| | | Translational elongation | 80/114 | < 0.001 | −2.12 | 126 |
| | PANTHER 2016 | Apoptosis signaling pathway | 43/102 | < 0.001 | −1.69 | 15.8 |
| | | Toll receptor signaling pathway | 23/49 | < 0.001 | −1.39 | 8.6 |
| | | T cell activation | 29/73 | < 0.001 | −1.39 | 7.1 |
| | Reactome 2016 | Eukaryotic translation elongation | 76/89 | < 0.001 | −2.10 | 163 |
| | | Peptide chain elongation | 72/84 | < 0.001 | −2.01 | 151 |
| | | Viral mRNA Translation | 70/84 | < 0.001 | −1.95 | 136 |
| 681 DETs for viral titer (q < 0.1) | GO Biological Process 2015 | Leukocyte activation | 38/373 | < 0.001 | −2.37 | 29 |
| | | Lymphocyte activation | 32/204 | < 0.001 | −2.32 | 24 |
| | | Regulation of leukocyte activation | 34/390 | < 0.001 | −2.53 | 19 |
| | PANTHER 2016 | Endothelin signaling pathway | 27/576 | 0.29 | −1.54 | 1.9 |
| | | T cell activation | 26/846 | 0.29 | −1.50 | 1.9 |
| | | JAK/STAT signaling | 42/808 | 0.28 | −1.44 | 1.8 |
| | Reactome 2016 | Adaptive immune system | 453/762 | < 0.001 | −2.28 | 17 |
| | | Infectious disease | 28/348 | 0.005 | −2.38 | 12 |
| | | HIV infection | 21/222 | 0.005 | −2.37 | 12 |

DET, differentially expressed transcript; FCH, fold change; GO, gene ontology; HIV, human immunodeficiency virus.
[a]Gene sets were constructed by thresholding DETs at the specified FCH and q-values and mapping to unique gene symbols.
[b]q-values are P-values adjusted using the Benjamini–Hochberg method, controlling the false discovery rate.
[c]Combined scores are the product of negative log P-values and the Z-score as described previously (Chen *et al*, 2013); the top three terms per annotation set, ordered by combined score, are displayed in this table.

At a threshold of FDR < 0.05, two of these coEMs were significantly enriched for at least one of five gene sets derived from the previously acquired DET signatures for timepoint and viral titer (Fig 7B), specifically turquoise (5/5 sets; max FDR $P = 9.6 \times 10^{-19}$; Fisher's exact) and sienna (3/5 sets; max FDR $P = 0.0022$). To explore the co-regulatory structure between coEMs and clinical variables, we correlated each coEM eigengene (which is the first principal component of the expression of genes in the module) against all other coEM eigengenes and the clinical variables (Fig 7C). This revealed that the turquoise module was strongly positively correlated with the convalescent phase (Pearson's $r = 0.82$), while the sienna module was very strongly positively correlated with the greenyellow module ($r = 0.97$) and negatively correlated with the convalescent phase ($r = -0.39$) and turquoise module ($r = -0.40$). This suggests that in our study, the sienna and greenyellow modules are most representative of acute-associated genes, while the turquoise module is most representative of convalescent-associated genes.

We again utilized gene set enrichment analysis to explore the composition of these modules. The sienna module was most significantly enriched for GO, Reactome, and WikiPathways terms regarding the regulation of cytokine production, immune system signaling, and type II IFN signaling (Table 3). On the other hand, the

greenyellow module did not achieve any significant enrichments among these annotation libraries at FDR < 0.1, and the size of the turquoise module (10,589 genes) precluded meaningful enrichment analysis.

To create a multiscale model spanning all experimental measurements, we expanded the interaction network to include correlations with cell sub-community frequencies (from CyTOF) and serum cytokine concentrations (from Luminex). When adding the latter dataset, the large positive correlations between nearly all cytokine concentrations mentioned previously (Appendix Figs S13–S15) created two large, interconnected components dominating the network structure (small black nodes, Appendix Fig S27) that clustered relatively far from all of the clinical variables. Focusing on only transcriptomic modules and cell sub-community data and restricting to Pearson correlations significant at $P < 0.001$, a smaller, well-organized network formed around the primary contrast in our dataset, the acute vs. convalescent timepoints (Fig 7D). Under a force-directed layout, cell sub-communities and gene modules that positively correlate with the convalescent timepoint clustered together (dashed gray box, Fig 7D), while cell sub-communities and gene modules that negatively correlate with convalescence (and therefore associate with acute infection) also

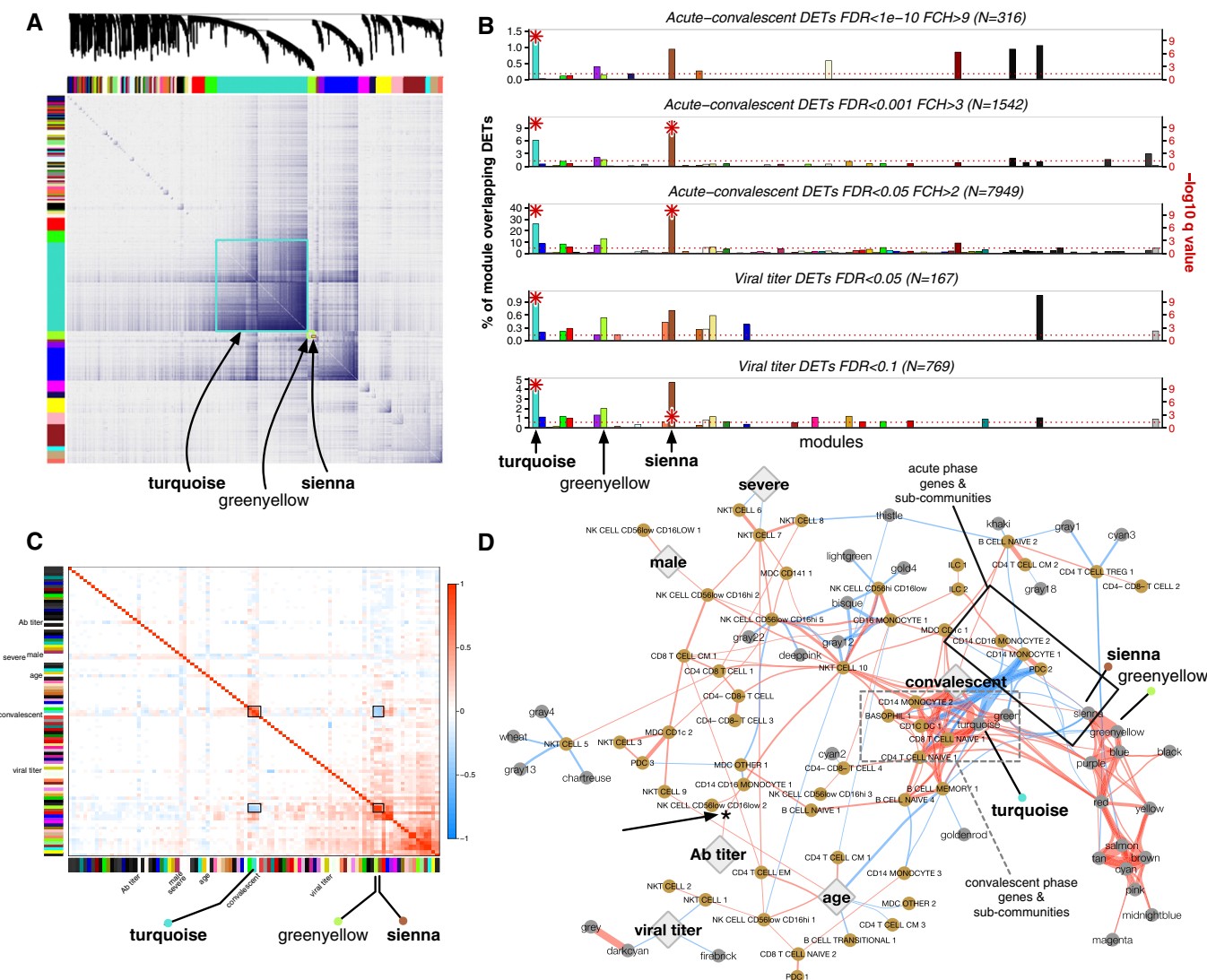

**Figure 7. A multiscale interaction network of gene modules and cell sub-communities for the human immune response to CHIKV.**

A  Topological overlap matrix (TOM) plot of coexpression network created from gene expression profiling of all samples across both timepoints (n = 42 patients × 2 timepoints × 2 technical replicates = 168 samples). At top, dendrogram of the hierarchical clustering of the matrix that undergoes a dynamic tree-cut operation to form 92 gene coexpression network modules (coEMs), depicted by the colored bars on the edges of the TOM plot. Color assignment is arbitrary. Three coEMs are highlighted (boxes).

B  Enrichment of five subsets of the DET signatures for CHIKV infection phase and viral titer (see Fig 6A and C) among each of the 92 coEMs (X-axis), showing the fractional overlap of the module with the DET signature (Y-axis). Negative log10-scaled q-values (Benjamini–Hochberg-adjusted P-values; Fisher's exact test) are depicted as red asterisks on the secondary Y-axis (right-hand side), clipped to a maximum of 10. The only modules with overlaps in any of the subsets that exceed FDR < 0.05 are turquoise and sienna. The horizontal line indicates a threshold of FDR < 0.05. The three highlighted coEMs from (A) are again labeled.

C  Correlations between coEM eigengenes (colored bars on each axis) and the clinical variables (text labels; Ab, anti-CHIKV antibody). The three coEMs in (A) are again highlighted. There is a strong inverse relationship between turquoise + convalescent vs. greenyellow + sienna.

D  Multiscale network of cell sub-community and coEM eigengenes depicted with a force-directed layout and edge bundling. Gold nodes, cell sub-communities; gray nodes, coEM eigengenes; large diamonds, clinical variables; red edges, positive correlations; blue edges, negative correlations. Edges are filtered to Pearson correlations significant at P < 0.001, and thickness corresponds to the square of the correlations. A cluster of sub-communities and coEMs associated with convalescence (dashed gray box) and a corresponding cluster of sub-communities and coEMs associated with acute infection (solid black box) surround the "convalescent" node. A positive correlation between convalescent anti-CHIKV antibody (Ab) titer and CD14+CD16+ monocyte sub-community 1 frequency (shown previously in Appendix Fig S3A and B) is also visible (asterisk and arrow).

clustered together (solid black box, Fig 7D). Weaker correlations between other gene modules, cell sub-communities, and the other clinical variables remained on the periphery of the network. For instance, the previously described significantly positive correlations between "nonclassical" CD14+CD16++ monocytes (sub-community 1) and convalescent anti-CHIKV antibody titer (Appendix Fig S3) reappears here as a weakly positive correlation (asterisk with arrow, Fig 7D).

**Table 3. Gene set enrichment analysis of coexpression modules.**

| Module (# genes[a]) | Annotation set | Term | Overlap | q-value[b] | Z-score | Combined score[c] |
|---|---|---|---|---|---|---|
| Sienna (370) | GO Biological Process 2015 | Regulation of cytokine production | 28/482 | 0.000268 | −2.51 | 20.7 |
| | | Positive regulation of cytokine production | 22/327 | 0.000279 | −2.45 | 20.1 |
| | | Regulation of immune effector process | 18/264 | 0.00106 | −2.46 | 16.9 |
| | Reactome 2016 | Immune system | 65/1547 | 1.81e-07 | −2.23 | 34.7 |
| | | Cytokine signaling in immune system | 26/620 | 0.0211 | −2.39 | 9.21 |
| | | Hemostasis | 23/552 | 0.0340 | −2.12 | 7.16 |
| | WikiPathways 2016 | Type II interferon signaling (IFNG) | 6/37 | 5.35e-05 | −1.82 | 10.4 |
| | | BDNF signaling pathway | 9/144 | 0.00141 | −1.92 | 6.84 |
| | | Senescence and autophagy in cancer | 8/105 | 0.000720 | −1.77 | 6.72 |
| Greenyellow (507) | GO Biological Process 2015 | Negative regulation of smooth muscle cell migration | 4/12 | 0.223 | −2.65 | 3.97 |
| | | Regulation of smooth muscle cell migration | 6/31 | 0.223 | −2.51 | 3.76 |
| | | Endosome to lysosome transport | 5/31 | 0.662 | −2.61 | 1.08 |
| | Reactome 2016 | TNF signaling | 5/41 | 0.459 | −2.08 | 1.62 |
| | | Deposition of new CENPA-containing nucleosomes at the centromere | 5/52 | 0.459 | −2.04 | 1.59 |
| | | Nucleosome assembly | 5/52 | 0.459 | −2.03 | 1.58 |
| | WikiPathways 2016 | Apoptosis modulation and signaling | 8/93 | 0.353 | −2.09 | 2.18 |
| | | Complement and coagulation cascades | 6/59 | 0.353 | −1.90 | 1.98 |
| | | Apoptosis modulation by HSP70 | 3/19 | 0.412 | −1.49 | 1.32 |

GO, gene ontology; IFNG, interferon-gamma; BDNF, brain-derived neurotrophic factor; TNF, tumor necrosis factor; CENPA, centromere protein A; HSP70, 70-kilodalton heat shock protein.
[a]Number of unique gene symbols.
[b]q-values are *P*-values adjusted using the Benjamini–Hochberg method.
[c]Combined scores are the product of negative log *P*-values and the Z-score as described previously (Chen *et al*, 2013); the top three terms per annotation set, ordered by combined score, are displayed here.

To test whether the severe dimensionality reduction of the three datasets to the relatively small network model presented in Fig 7D retained predictive value for the acute-convalescent contrast, we fit elastic net regularized logistic regression models to three versions of the merged data: *A*, complete per-transcript quantification, sub-community frequencies, and serum cytokine concentrations; *B*, same as *A* but with coEM eigengenes instead of per-transcript quantification; and *C*, same as *B* but with serum cytokine concentrations removed, as in the final network model. As expected, given the strong transcriptomic signature for time-point, the model that had access to complete per-transcript quantification achieved nearly perfect performance under fivefold cross-validation (area under the receiver operating characteristic [AUC] = 0.98, 95% confidence interval [CI] 0.93–1.00; Appendix Fig S28A). Replacing per-transcript quantification with the 92 coEM eigengenes actually slightly increased predictive performance (AUC = 0.99, 95% CI: 0.96–1.00; Appendix Fig S28B). Removing the serum cytokine data, leaving only the dimensionality-reduced data used to generate Fig 7D, did not noticeably detract from model performance (AUC = 0.97, 95% CI: 0.91–1.00; Appendix Fig S28C). This indicates that the compact, multiscale network model presented in Fig 7D delivers nearly optimal predictive value for the timepoint contrast despite the > 1,000-fold reduction in dimensionality from the original dataset.

# Discussion

In this study, we present comprehensive immune profiles for natural human infection with CHIKV. By employing a diverse CyTOF antibody panel and novel clustering techniques to systematically discover sub-communities and their frequencies in these data, we discovered more heterogeneity within PBMC populations than previously recognized in previous studies of viral infection, including identification of a novel monocyte sub-community. Considering the scarcity of CyTOF data from viral infections (Miner *et al*, 2015; Sen *et al*, 2015), these results may represent the most comprehensive immune profiles for any human viral disease to date. Further, by taking an unbiased approach in measuring global changes across three scales (cell subpopulations, gene expression, and serum cytokines), we provide unprecedented robustness and detail on the effects of CHIKV infection on humans and a number of novel findings that suggest future hypothesis-driven studies.

## Strong associations between CHIKV and monocyte sub-communities

On the cell population level, our findings indicate a prominent role for CD14[+] and CD14[+]CD16[+] monocytes—including several novel phenotypes therein—during the acute immune response to CHIKV.

Two cell sub-communities were more strongly associated with acute infection than all other sub-communities by two orders of magnitude: "intermediate" $CD14^{++}CD16^{+}$ monocytes and a $CD123^{+}$, $CX_3CR1^{+}$, and $CD141^{+}$ subpopulation of $CD14^{+}$ monocytes.

"Intermediate" $CD14^{++}CD16^{+}$ monocytes are a recently described population (Ziegler-Heitbrock et al, 2010; Wong et al, 2011) that received initial attention for showing independent predictive value for cardiovascular event risk (Rogacev et al, 2012) that hinted at a role in vascular inflammation or atherosclerosis. Unlike "nonclassical" $CD14^{+}CD16^{++}$ monocytes, these cells selectively express CCR5 (Ellery et al, 2007) and are expanded in bacterial sepsis, dengue, Crohn's disease, rheumatoid arthritis, Eales' disease, and asthma (Wong et al, 2012). Compared to our putative cluster of "nonclassical" $CD14^{+}CD16^{++}$ monocytes, we too saw higher expression of CCR5 in the "intermediate" subpopulation, as well as higher expression of HLA-DR, another selective marker (Appendix Fig S4), providing strong evidence that this population is the same "intermediate" phenotype described in previous studies (Zawada et al, 2011). In vitro studies were able to induce the "intermediate" phenotype from $CD14^{+}CD16^{-}$ monocytes by treatment with IL-10 (Tsukamoto et al, 2017), a cytokine that we also found was elevated during acute CHIKV infection (Fig 5E). Since "intermediate" monocytes are known to be potent secretors of IL-10 (Skrzeczyńska-Moncznik et al, 2008), a positive feedback loop involving IL-10 could contribute to the strong, selective expansion in "intermediate" monocytes that was observed during the acute phase. Furthermore, intermediate monocytes secrete many other cytokines and are potent phagocytic cells during infection (Fingerle et al, 1993; Ziegler-Heitbrock & Hofer, 2013). Given how much remains to be characterized about "intermediate" monocytes, and since they associate with inflammatory and autoimmune joint diseases that resemble chronic CHIKV arthropathy (like rheumatoid arthritis), this is an exciting avenue for further inquiry (Chaaitanya et al, 2011; Nakaya et al, 2012; Miner et al, 2015).

Additionally, we discovered a novel subpopulation of $CD14^{+}$ monocytes associating with the acute phase of infection that expressed high levels of CCR4, CXCR3, and CCR6, among other markers (Appendix Fig S8). Although it was a rare population, comprising approximately 1% of all monocytes, these markers have never been described in association with a distinct subpopulation of monocytes and are instead more classically attributed to T cells. Manual re-gating and cross-sample comparisons confirmed that this sub-community is not an artifact of our semi-supervised clustering approach (Appendix Figs S7 and S11). This demonstrates that the heterogeneity of monocytes may extend beyond the "nonclassical", "intermediate", and "classical" divisions (Appleby et al, 2013) and that more careful identification of specific subpopulations—perhaps not only by protein expression but also transcriptomic profiling—will be useful for fully understanding the diversity of monocyte functionality (Stansfield & Ingram, 2015).

A globally significant correlation was found between "nonclassical" $CD14^{+}CD16^{++}$ monocyte frequency in the acute phase and the anti-CHIKV antibody titer 2 weeks later (Appendix Fig S3). This suggests that this subpopulation could contribute to the development of a stronger humoral response, with potentially long-term implications, given an association discovered between the early antibody response and decreased likelihood of chronic arthralgia (Kam et al, 2012). Recent studies have suggested that monocytes

can play a role in modulating activation of certain T cells (Eberl et al, 2009; Charron et al, 2015), which implies that monocyte subpopulations may contribute to the success of the adaptive immune response for certain pathogens. One study of this crosstalk reported induction of cytokines in δγ T cell–monocyte co-culture that closely matched the pattern of upregulation seen in our study, including IFN-γ, TNF-α, CCL2, IL-8, and CXCL10 (Eberl et al, 2009). Even more relevant is a recent study of infection of $CD14^{+}$ monocytes by DENV, which upregulated CD16 expression and induced differentiation of B cells into plasmablasts, ultimately increasing IgG secretion (Kwissa et al, 2014). Considering that this sequence of events closely mirrors the findings of our study, such as the high expression of a CHIKV envelope protein in monocytes, upregulation of $CD14^{+}CD16^{+}$ monocytes during acute infection, and the correlation with convalescent-phase anti-CHIKV antibody titers, a similar mechanism may apply equally to CHIKV infection.

### CHIKV envelope protein is expressed by specific B cell sub-communities

Interestingly, we discovered two distinct sub-communities of B cells that significantly expressed CHIKV E protein during acute infection: memory B cells and a sub-community of naïve B cells (sub-community 4; see Fig 3E). Within the four sub-communities of naïve B cells, two of them (labeled 1 and 4) were very similar in regard to their pattern of marker expression of CXCR5, CCR4, CXCR3, and CD80 (Appendix Fig S2). In particular, CXCR5 expression was very high on sub-community 4, which together with CCR7 is an important chemokine receptor for the migration of lymphocytes to secondary lymphoid tissue and is important for follicle formation within tissue (Stein & Nombela-Arrieta, 2005). The expression of CCR4, CXCR3, and CD80 was lower on sub-community 4 compared to the other sub-communities. Based on previous studies, CXCR3 is absent in normal naïve B cells but is upregulated on malignant B cells while CD80 plays an important role for the activation of autoreactive T cells during rheumatoid arthritis (Trentin et al, 1999; O'Neill et al, 2007). In the light of these studies, it appears that the different type of naïve B cells we detect by CyTOF reflect different degrees of activation, with sub-communities 1 and 4 being more activated based on their higher expression of CCR4, CXCR3, and CD8, suggesting they may be in the process of transforming into memory B cells.

### Serum cytokines support a monocyte-centric response to CHIKV

Most of the cytokines that were increased during acute infection were likely secreted by monocytes or contributed to the observed expansions of monocyte subpopulations (Fig 5). The pronounced acute-phase increase in CXCL10 is likely linked to secretion by monocyte populations that expanded during the acute phase, since both monocytes and T cells secrete CXCL10 (Luster & Ravetch, 1987). CXCL10, which induces chemotaxis in monocytes, monocyte-derived cells, and T cells, can also be induced by IFN-α, IFN-γ, and TNF-α, which were all upregulated in our study during the acute phase of CHIKV infection (Liu et al, 2011). Changes in CXCL10 expression are well correlated with many infectious diseases (Liu et al, 2011), although the role of CXCL10 in viral pathogenesis and its signaling pathways is still poorly understood

(as it seems to alternately promote or protect against infection in different studies).

We propose that a feed-forward loop between production of IL-10 by "intermediate" $CD14^{++}CD16^+$ monocytes (Skrzeczyńska-Moncznik et al, 2008) and induction of the "intermediate" phenotype by IL-10 (Tsukamoto et al, 2017) could help explain the upregulation of IL-10 during acute CHIKV infection. Another contribution could be from T cells, since monocyte–T cell interactions are able to activate T cells to an IL-10-secreting state (Charron et al, 2015). IL-10 secretion is known to be dependent on a number of regulatory factors that were transcriptionally upregulated in our study, including p38 and MyD88 (Appendix Fig S21; Saraiva & O'Garra, 2010). Another cytokine that likely contributed to the observed expansions in $CD14^+$ and $CD14^+CD16^+$ monocyte subpopulations is CCL2, a known $CD14^+$ monocyte chemoattractant (Serbina et al, 2008), which we observed to be upregulated during acute infection. CCL2 is secreted by monocytes and fibroblasts among many other cell types (Van Damme et al, 1994), and like IL-10, its secretion from monocytes is also p38-dependent (Fietta et al, 2002).

CCL2 was the only cytokine that positively correlated with all monocyte subpopulation frequencies during the convalescent timepoint, which significantly differed from the correlations observed for all other cytokines (Appendix Fig S16). Finally, although we did not see significant upregulation of serum CCL7 levels, and the antibody panel did not include CCL8, RNA-seq did reveal that both of these genes were in fact transcriptionally upregulated during the acute phase (Appendix Fig S18). These gene products are both CCR2-binding monocyte chemoattractants (although less well characterized than CCL2) and therefore may have also contributed to expansions in monocyte populations.

Previous studies that profiled immunological changes associated with CHIKV infection in humans focused on serum cytokine levels (Ng et al, 2009; Chaaitanya et al, 2011; Chow et al, 2011; Schilte et al, 2013; Teng et al, 2015). Our results were largely concordant, but there were also some notable differences. Of the significantly elevated cytokines and growth factors in our data, CCL2, CCL4, CXCL10, G-CSF, IFN-α, IFN-γ, IL-1Ra, IL-2, IL-6, IL-10, IL-12p40, and IL-15 concur with the changes supported by a recent systematic meta-analysis of these studies (Teng et al, 2015). Owing perhaps to the breadth of our cytokine panel—or the unique focus on pediatric cases—we observed many changes that were not found in the meta-analysis. In our study, very significant increases in CXCL8 were observed during the acute phase, although previous studies varied on whether it was upregulated or downregulated in the acute phase, and this may be due to a dependency of the effect on the amount of virus in the bloodstream (Teng et al, 2015). We observed significant increases in TNF-α, CCL5 (aka RANTES), and IL-1α, all of which were measured infrequently in previous studies (Teng et al, 2015), although CCL5 was reported by one study as a potential biomarker for severity (Ng et al, 2009). We also uniquely observed increases in CCL11, GM-CSF, TGF-α, and $CX_3CL1$, which were not measured by enough previous studies to be included in the meta-analysis. Finally, most interestingly, we report a decrease in IL-17A in our patients during acute infection (the only significantly decreased cytokine; see Fig 5) while it had been consistently upregulated in previous studies (Teng et al, 2015). Altogether, reviews have already noted considerable variability in the results for serum

cytokine studies on CHIKV (Burt et al, 2017), suggesting that either much larger cohorts or a wider variety of profiling data will be needed to generate more reproducible profiles. In part, this was a motivating factor for us to incorporate cell subpopulation and transcriptomic data into our immune profiles of CHIKV infection.

### Transcriptomic signatures for CHIKV infection phase, viremia, severity, and immunogenicity

Overall, our results are consistent with the transcriptomic signature recently reported for a C57BL/6J mouse model of CHIKV infection (Wilson et al, 2017). We both find that strong acute-phase transcriptional upregulation occurs in many interferon-associated genes, including IFIs, MX1 and MX2, OAS genes, and RSAD2 (Viperin) (Wilson et al, 2017). Notably, in mice, CXCL10 and CXCL9 were the most upregulated cytokine genes when comparing the acute phase to controls, with CCL2 also substantially upregulated (Wilson et al, 2017), mirroring our measurements of the most significantly modulated serum cytokine concentrations. These data show some consistency to the monocyte-centric immune response to CHIKV between mice and humans. Our finding that type I IFN genes are not upregulated during acute infection—while initially surprising since serum concentrations of IFN-α were in fact elevated—turns out to be consistent with the mouse model, which found very low RNA abundance for type I IFN transcripts (Wilson et al, 2017). Likewise, among IFN-regulated transcription factors, we find a concordant pattern of IRF7 upregulation during the acute phase, while IRF3 is not upregulated (Appendix Fig S23). Together, these data establish that this mouse model of CHIKV replicates many aspects of the gene expression signature induced by CHIKV in humans, including modulations of monocyte-related cytokines, and therefore, our data support its continued use as a model of human CHIKV infection and the corresponding immune response.

Besides a large acute-convalescent transcriptomic signature, we were also able to elucidate three novel transcriptomic signatures for variance in CHIKV viral titer, symptom severity and the 15-day p.s.o. anti-CHIKV antibody titer. Notably, these signatures were constructed after incorporating timepoint as a covariate—that is, they were found to significantly add information to a model that had already accounted for timepoint, age, and gender. Of these signatures, the strongest and least surprising was the signature for higher acute-phase CHIKV viral titer, which was enriched for leukocyte activation, adaptive immune system, and interferon signaling genes. The presence of a distinct signature for viral titer in our data indicates that a more viremic acute phase must have led to transcriptional upregulation of these genes across both timepoints; otherwise, they would be sufficiently captured by the acute-convalescent DET signature alone.

The transcriptional signatures for symptom severity and immunogenicity were notable for having an abundance of HLA (aka major histocompatibility complex [MHC]) transcripts—in particular, HLA-A, HLA-B, HLA-DMA, and HLA-DMB. Our study showed that certain HLA-B transcripts appeared to be associated with less severe acute-phase symptoms (Fig 6D), while certain HLA-A, HLA-DMA, and HLA-DMB transcripts were correlated with changes in the convalescent anti-CHIKV antibody titer (Fig 6F). It is not surprising that HLA gene expression could affect infection outcomes, since the HLAs are responsible for antigen presentation and both the adaptive

and cellular immune responses. As IFN-$\alpha$ and IFN-$\gamma$ increase the transcription of *HLA-B* among other MHC class I loci during acute viral infection (Girdlestone, 1995), a particularly strong *HLA-B* response could boost the cytotoxic immune response and mitigate acute-phase symptoms. Based on known association between acute viral infection and the MHC region, it is reasonable to infer that more transcription of *HLA-DMA* and *HLA-DMB* (class II alleles) would correlate with higher anti-CHIKV antibody titers (Fig 6F), since they are involved in both the adaptive and humoral responses. Our data, therefore, could be interpreted to reflect different relative prioritization of these immune responses among our patients that manifests as a difference in magnitude of the early antibody response, which was previously observed to correlate with decreased risk of chronic arthralgia (Kam *et al*, 2012). Since allelic diversity in HLA loci is well established as a genetic risk factor for certain infectious and autoimmune diseases, our signatures may also reflect underlying genetic variation (e.g., expression quantitative trait loci [eQTL]) that affects transcription of HLA genes and thereby shifts disease outcomes (Kumar *et al*, 2014). Finding host genetic factors for CHIKV severity could lead to further insight into mechanisms of pathogenesis, so this is a promising direction for future study.

### A multiscale network model of CHIKV pathogenesis

Finally, our study generated a network model that integrates global measurements of cell sub-communities, cytokines, and gene transcription into a compact roadmap of the immune responses triggered by CHIKV (Fig 7D). A network that leverages modularity is valuable because of the inherent limitations of gene-level or cell-level analyses, which are poorly suited for traditional inference testing or Bayesian analysis because of their high dimensionality and non-independence among many of the observations. To our knowledge, this is the first attempt to combine WGCNA for detecting transcriptional network modularity with comprehensive cell sub-community frequencies modeled within CyTOF data.

Whole-genome coexpression network analysis produced 92 coexpression gene modules, two of which were significantly enriched for DET signatures for either infection phase or viral titer. One of these coEMs, sienna (426 genes), was significantly enriched for cytokine signaling and immune signaling terms and correlated with the acute phase of infection; a second much larger module, turquoise (10,589 genes), strongly correlated with the convalescent phase of infection. Combining gene modules with subpopulation frequencies and serum cytokine concentrations into a correlational network and filtering for edges at $P < 0.001$ produced a network dominated by intracorrelation in the cytokines (Appendix Fig S27). Since none of the cytokines correlate significantly with the clinical variables, we removed them from the network to produce a more compact model (Fig 7D). Under a force-directed layout, this network organizes around the primary contrast in our data—the phase of infection—with acute-phase vs. convalescent-phase genes and cell subpopulations separating into two communities. The sienna module also serves as a central "bridge" between the timepoint contrast and most of the other strong interactions between gene modules in our dataset.

Although limited by sample size and the specific timepoints used in our study, this network represents the first completely data-driven model of the immune reaction to CHIKV across multiple layers of "omics" data. It compactly summarizes changes of hundreds of thousands of measured analytes with essentially no reduction in the predictive value for the timepoint contrast (Appendix Fig S28) and puts these interactions into global context with other clinical variables. We hope that the generation of similar multiscale networks for other viral infections, e.g., DENV and ZIKV, will soon lend insight into the comparative effects of these viruses on the human immune system and aid in the discovery of therapeutics and prognostic biomarkers that remain robust across the multiplicity of arboviral infections now prevalent in tropical urban regions.

In conclusion, our comprehensive immune profiling of 42 pediatric cases of CHIKV infection revealed an immune response largely centered on changes in monocyte subpopulations and monocyte-related cytokines. Monocytes displayed the highest change in CHIKV E2 protein expression between the two timepoints. An "intermediate" CD14$^{++}$CD16$^{+}$ subpopulation and an activated (CD123$^{+}$, CX$_3$CR1$^{+}$, and CD141$^{+}$) CD14$^{+}$ monocyte subpopulation associated most strongly with the acute phase of infection when compared against all other identified subpopulations of PBMCs. Interestingly, we also found a small subpopulation of CD14$^{+}$ monocytes with distinctly higher expression of previously unreported markers (CCR4, CXCR3, and CCR6) that also associated with the acute phase of infection. Although "nonclassical" CD14$^{+}$CD16$^{++}$ monocyte frequencies were unchanged across the timepoints, we found a significant correlation between their frequency in both phases and the corresponding convalescent-phase anti-CHIKV antibody titers. Finally, among the elevated serum cytokine levels for the acute phase of infection, many were monocyte chemoattractants or secreted by monocytes and macrophages (e.g., CXCL10, CCL2, and IL-10), as described previously (Duque & Descoteaux, 2014; Boyette *et al*, 2017).

Our study produced additional novel findings. We confirmed that transcriptomic effects for the different phases of CHIKV infection in humans correspond well to those recently reported for a mouse model (Wilson *et al*, 2017), but furthermore, we discovered new transcriptomic signatures for the level of acute-phase viremia, acute-phase symptom severity, and convalescent-phase immunogenicity. Among the signatures for acute severity and convalescent immunogenicity, we found an abundance of specific *HLA* transcripts that correlated with both of these outcomes, and a notably strong correlation between severity and transcription of *MXRA7*, an essentially uncharacterized gene with only two unrelated disease associations reported in the literature (Veiga-Castelli *et al*, 2010; Sim *et al*, 2013). We also find globally significant expression of CHIKV E2 protein on two B cell subpopulations, which have never been productively infected *in vitro* by CHIKV nor detected in infected patient samples (Sourisseau *et al*, 2007; Her *et al*, 2010; Teng *et al*, 2012). Finally, we have integrated all of our observed changes into a multiscale network that summarizes the immunological changes across the cellular and gene expression levels and their interactions with certain clinical outcomes. We believe that our findings have provided a uniquely global perspective on the biomolecular and immunological landscape of CHIKV infection and that they spark new hypotheses for future experiments that can further disentangle the mechanisms of CHIKV pathogenesis and the components of a successful immune response in humans.

# Materials and Methods

## Study participants

To characterize the immune profiles in CHIKV infection, 43 chikungunya cases (42 for analysis plus one extra) were selected from participants aged 6 months to 14 years who were enrolled in our ongoing study at the National Pediatric Reference Hospital (Hospital Infantil Manuel de Jesús Rivera; HIMJR) in Managua, Nicaragua, between September 2015 and April 2016. Chikungunya cases were laboratory-confirmed by detection of CHIKV using real-time RT–PCR (Waggoner *et al*, 2016), in some cases followed by virus isolation, in acute-phase samples. In addition, seroconversion by IgM capture ELISA and/or a >4-fold increase in antibody titer by Inhibition ELISA in paired acute and convalescent sera were evaluated (Galo *et al*, 2017). Participants were also screened for dengue virus (DENV) infection, and CHIKV/DENV co-infections were excluded. Furthermore, children with severe symptomology were excluded to obtain a more homogenous set of patient samples. All selected cases had an acute-phase sample collected on days 1–2 of illness and a convalescent sample collected on days 15–17 post-illness for serum, whole blood in PAXgene solution, and PBMCs; these were subject to transcriptomic analysis via RNA-seq, CHIKV viral titer assays, multiplex ELISA for cytokines, and mass cytometry for cellular phenotyping as illustrated in Fig 1. PBMCs were isolated from whole blood as previously described (Zompi *et al*, 2012). Additionally, follow-up samples were collected at 3, 6, and 12 months post-infection to study long-term outcomes of CHIKV infection. Sampling times closely adhered to the targeted acute (standard deviation [SD] = 0.5 days) and convalescent (SD = 0.6 days) timepoints. Clinical information was collected every 12 h, and after systematic monitoring by a clinical supervisor was digitized by double-data entry with quality control checks performed daily and weekly. This study was conducted as a collaboration between the Nicaraguan Ministry of Health and the University of California, Berkeley, and was reviewed and approved by the Institutional Review Boards (IRBs) of the University of California, Berkeley, and the Nicaraguan Ministry of Health. Parents or legal guardians of all subjects provided written informed consent, and subjects 6 years of age and older provided verbal assent as approved by the IRBs.

## CyTOF sample processing and acquisition

Cytometry time-of-flight uses metal-labeled reagents and inductively coupled plasma mass spectrometry to overcome the limits of fluorescence spectral overlap in flow cytometry, allowing measurement of up to 50 analytes at single-cell resolution. Cryopreserved PBMC samples from the acute and convalescent phases of infection were thawed and stained with Rh103 nucleic acid intercalator (Fluidigm) as a viability marker. Paired PBMC samples from each timepoint were first barcoded using a CD45 antibody-based barcoding approach (Mei *et al*, 2016), and each acute and convalescent sample pair was pooled as a single patient sample for subsequent processing to minimize technical variability and potential batch effects. The pooled patient samples were then stained with a validated 37-marker CyTOF antibody panel (Table EV8) for 30 min on ice and then fixed, permeabilized, and incubated with Ir nucleic acid intercalator (Fluidigm). The samples were then stored in freshly diluted

2% formaldehyde in PBS and stored until acquisition. Immediately prior to CyTOF acquisition, the samples were washed with deionized water (diH20), counted, and resuspended in diH20 containing a 1/20 dilution of Eq 4 Element beads (Fluidigm). Following routine autotuning, the samples were acquired on a CyTOF2 mass cytometer (Fluidigm) equipped with a SuperSampler fluidics system (Victorian Airships) at an event rate of <400 Hz.

## CyTOF data analysis

Following data acquisition, the FCS files were normalized using the bead-based normalization algorithm in the CyTOF control software and uploaded to Cytobank for initial data processing. Normalization beads were excluded based on Ce140 signal, and cell events were identified based on Ir191/193 DNA signal. A conservative doublet exclusion gate was applied based on DNA and event length, and Rh103$^+$ dead cells were also excluded. The cell events associated with the acute and convalescent samples were then manually de-barcoded based on CD45-194Pt and CD45-198Pt expression, respectively, and were split and exported as separate samples for subsequent analyses using a semi-supervised computational analysis pipeline. Potential intra-individual batch effects were minimized by pairing samples for the same individual into the same batch and assessed by looking at three candidate batch variables: (i) acquisition, (ii) staining, and (iii) thawing dates for each sample (Appendix Fig S29). Using the first principal component of the CyTOF data, which captures 78% of the variance, we evaluated the contribution of those three variables to the variance and found no association (all show $P > 0.7$). To quantify how much variability in the data is explained by the batch variables, we used principal variance component analysis (PVCA). Since the 3 potential sources of batch effect were largely collinear (Spearman correlation > 0.93 for thawing, staining, and acquisition batch variables), we included only the acquisition date in our PVCA analysis, which indicated that this variable explains only 3.8% of the overall variance (Appendix Fig S29). Since this contribution is much lower than that of the clinical data—such as patient ID, patient age, and timepoint—which collectively contribute to 79.5% of the observed variance, or approximately the size of the first principal component, we removed the "acquisition date" variable from downstream modeling.

Traditional hierarchical gating was applied to a subset of samples to identify 8 major immune compartments: T cells, B cells, NK cells, NKT cells, monocytes, mDCs, pDCs, and basophils. These manually gated data were used to train a logistic regression classifier (which we term Nod), which was then applied to identify these populations in all the samples; fitted model coefficients are provided in Table EV9. The logistic classifier is important for the stratification of major cell subsets prior to performing unbiased Louvain community detection. We tested the logistic regression model performance over the cell subsets, both by manually examining the prediction and by determining the F1 score (Appendix Fig S30). In general, our logistic regression model shows acceptable performance for all cell subsets as measured by both precision/recall (Appendix Fig S30A) and F1 score (Appendix Fig S30B), with larger subsets showing better performance due to the increased availability of training data. Next, we applied Phenograph (Levine *et al*, 2015) as an unbiased approach to define the phenotypic heterogeneity within each of

these compartments (HybridLouvain). The cell clusters identified in each single sample were then meta-clustered across all samples to identify phenotypically similar communities that were reproducibly present across multiple samples (MetaHybridLouvain). These meta-clusters were then manually annotated based on overall marker expression profiles and their association with known immune cell subsets, allowing for the presence of additional phenotypically distinct sub-clusters within these known subsets. These annotations were mapped back to the individual samples, and the relative frequency and median marker expression patterns of these consistently annotated clusters were then exported for further statistical analyses. Meta-clusters that were characterized by protein expression patterns that did not correspond to any known cell subsets, including those that appeared to be cell–cell doublets, were annotated as "undefined" and not included in subsequent statistical or multiscale network analyses.

## Multiplex ELISA

Cytokines and chemokines were measured using a multiplex ELISA-based assay (Luminex). All serum samples were inactivated with a UV-C lamp (254 nm) for 10 min on ice in a biosafety-level 3 laboratory at the University of California, Berkeley. Each sample was run in duplicate in a 96-well microtiter plate using 25 µl serum from each patient from acute and convalescent timepoints using the multiplex cytokine panels (Multiplex High Sensitivity Human Cytokine Panel, Millipore Corp.). Forty analytes (cytokines and chemokines) were measured using a Luminex-200 system and the XMap Platform (Luminex Corporation). Acquired mean fluorescence data were analyzed and calculated by the Beadview software. The lower and upper detection limits for these assays are 3.0 pg/ml and 15,000 pg/ml, respectively. Quality control of each sample was performed, and a bead count of < 50 was not used for analysis.

## Viral titer assays

Viral RNA was extracted from 140 µL of patient serum using the QIAamp Viral RNA Mini Kit (Qiagen) according to the manufacturer's protocol, and RNA was eluted in 60 µl of RNase-free water. Primers for the E1 gene were designed to quantify CHIKV copies in each patient and were used at 300 nM final concentration. The forward primer was 5′-CATCTGCACYCAAGTGTACCA-3′, and the reverse primer was 5′-GCGCATTTTGCCTTCGTAATG-3′. A TaqMan-labeled probe was used for detection: FAM-5′-GCGGTGTACACT GCCTGTGACYGC-3′-BHQ-1 (Waggoner *et al*, 2016). The SuperScript III One-Step RT-PCR System (Invitrogen) was used for reverse transcription of viral RNA and subsequent amplification of viral complementary DNA (cDNA). Specifically, 5 µl of extracted viral RNA, 0.5 µl of SuperScript III RT/Platinum Taq High Fidelity Enzyme Mix (Invitrogen), 12.5 µl of 2× Reaction buffer, 5 µl RNase-free water, and 2 µl of primers and probes were added to each well. Viral RNA was reverse-transcribed (52°C for 15 min), and the resulting cDNA was amplified via one cycle of denaturation (94°C for 2 min), 45 cycles of denaturation (94°C for 15 s), annealing (55°C for 40 s), and extension (68°C for 10 s). For quantification of CHIKV copies, a 4-point standard curve (8.0, 6.0, 4.0, and 2.0 $\log_{10}$ copies/µl of eluate) was used. Standard curves were prepared using quantitated ssDNA (Integrated DNA Technologies) containing the target

sequence of CHIKV with the following sequence: 5′-CACAACA TCTGCACCCAAGTGTACCACAAAAGTATCTCCAGGCGGTGTACACT GCCTGTGACCGCCATTGTGTCATCGTTGCATTACGAAGGCAAAATG CGCACTAC-3′.

## Preparation of RNA sequencing libraries

Total RNA was extracted from PAXgene RNA blood solution with the PAXgene Blood RNA Kit (Qiagen) by following the manufacturers' instructions including DNase digestion and an additional cleanup using RNeasy MinElute kit (Qiagen). Purified RNA samples were quantified by Qubit 3.0 fluorometer with RNA High Sensitivity Assay kit (Thermo Fisher). We confirmed the quality of the RNA with the RNA High Sensitivity ScreenTape using the TapeStation 2200 (Agilent Technologies). Sample libraries were then prepared from the 86 samples' libraries (from 42 paired patient samples and the 1 extra pair). First, ribosomal RNA (rRNA) and globin mRNA were removed from 200ng total RNA, and the remaining RNA was fragmented and primed for cDNA synthesis using TruSeq Total Stranded RNA HT kit with Ribo-Zero Globin on a Microlab STAR automated liquid handling system (Hamilton). The libraries were barcoded with TruSeq HT indices to allow for multiplexing, and ligation-mediated PCR was performed to enrich barcoded libraries for 15 cycles, and then purified with the Agencourt AMPure XP beads system (Beckman Coulter). The libraries were assessed for quality with the high-sensitivity DNA chip in a TapeStation 2200 (Agilent) and quantified with KAPA Library Quantification Kits for Illumina platforms (Kapa Biosystems). The libraries were diluted to 2 nM and combined equimolarly in pools of 12. These pools were then clustered using a cBot (Illumina) with a HiSeq 3000/4000 paired-end cluster kit on a patterned flow cell, one pool per lane. The flow cell was sequenced on a HiSeq 4000 using a HiSeq 3000/4000 SBS kit (300 cycles, Illumina). Two technical replicates were sequenced per biological sample, for a total of 168 sequencing runs.

## Pre-processing of RNA-seq data

Sequencer-generated base call (BCL) files were converted to FASTQ files, and the multiplexed samples were separated using bcl2fastq, which was then assessed for sequencing quality using FastQC (version 0.11.4, http://www.bioinformatics.babraham.ac.uk/projects/ fastqc/). The FASTQ files were quality filtered by using FASTX-Toolkit (http://hannonlab.cshl.edu/fastx_toolkit/) with the invocation fastq_quality_filter -q 30 -p 50 -v -Q 33, and only the sequencing reads that met all quality control requirements were aligned to the latest human reference genome (GRCh38) using HISAT2 (Kim *et al*, 2015; version 2.0.4). SAMtools (Li *et al*, 2009; version 0.1.19) was used to sort and convert the SAM files to BAM. Aligned sequences were assembled into potential transcripts, and gene expression in units of FPKM was quantified using StringTie (Pertea *et al*, 2015; version 1.2.2). Gene expression in units of overlapping read counts were also obtained by using the htseq-count script from HTSeq (Anders *et al*, 2015) on SAM files of pre-processed RNA-seq alignments.

## Differential expression analyses

For differential expression analysis at the transcript level, we used kallisto (Bray *et al*, 2016; version 0.43.0), a pseudo-alignment

method, and limma (Ritchie *et al*, 2015; version 3.30.6). Pseudo-alignment utilized a transcriptome index built from Ensembl release 79 (March 2015) for GRCh38. Transcripts where pseudo-alignment counts in units of counts per million (CPM) were < 1 in > 10 samples were removed, and the remaining 53,971 transcripts were analyzed.

Transcripts per million (TPM) values for transcript quantification and overlapping read counts (from HTSeq) for gene-level quantification were normalized using the voom methodology, which models the variance based on abundance and heterogeneity of the samples and typically achieves better control over FDR than other RNA-seq methods (Law *et al*, 2014). Additionally, it converts the data to a linear, normal scale allowing the use of classical linear models, including addition of covariates and extensions to models for longitudinal data.

Expression profiles were modeled using mixed-effects models in the limma framework to account for the paired structure of the data. limma uses an empirical Bayes moderation of the standard errors toward the prior transcript variances, which was fitted using an intensity-dependent trend. All models included age and gender as covariates. *P*-values from the moderated (paired) *t*-test were adjusted for multiple hypotheses using the Benjamini–Hochberg (FDR-controlling) procedure (Ritchie *et al*, 2015). For the acute vs. convalescent (timepoint) signature, age and gender were included as covariates in the model. For viral titer, severity, and convalescent anti-CHIKV antibody titer signatures, the covariates included in the model were timepoint, age, and gender. Viral titers and convalescent anti-CHIKV antibody titers were modeled as continuous variables in units of $\log_{10}$ dilutions. Pathway-based visualization of differentially expressed genes was performed with the pathview (Luo & Brouwer, 2013) R package and KEGG (Ogata *et al*, 1999) annotations.

## Gene expression analysis of cell composition in whole blood

Cell sub-community shifts derived from the CyTOF analysis were compared to changes in leukocyte composition estimated from differentially expressed genes in the RNA-seq data. We used computational methods to estimate cell components based on gene expression profiles of admixtures. Cell proportion estimation methodologies can be categorized into two main groups, based on whether it relies on known cell subset-specific marker genes or reference signature expression profiles of different cell subsets. For the former approach, we utilized cell markers of six major blood cell types obtained from the *IRIS* project and estimated abundance of each cell type by the average expression of its markers (Abbas *et al*, 2005). For the latter, we used the recently developed algorithm *CIBERSORT*, which requires an input matrix of reference gene expression signatures of different cell types (Newman *et al*, 2015). The *CIBERSORT* R package and its associated leukocyte signature matrix of 22 cell types were utilized with all default parameters.

Finally, we aimed to derive differentially expressed genes that are not purely caused by cellular component changes. To achieve this, we adjusted each gene's expression according to cellular abundance. Due to the relatively small sample size, we only considered the abundance of six main types of blood cells estimated by the average expression of cell markers (as described above): B cells, T cells, NK cells, monocytes, neutrophils, and dendritic cells.

Specifically, we used linear regression to obtain the residual gene expression after considering the cell abundance of the six major cell types in the model. We then used paired *t*-tests to obtain differentially expressed genes between the acute and convalescent phases. In this way, we obtained only 132 differentially expressed genes with a nominal *P*-value < 0.05, and none of them remained significant at FDR < 0.05 after multiple hypothesis corrections. The lack of significantly differentially expressed genes after adjusting for cell abundance suggested that the main signal of differential gene expression in whole blood is strongly derived from changes in cell subpopulation frequencies.

## Construction of gene coexpression networks and coexpression modules

Gene coexpression networks were constructed from the gene-level expression data for all samples using the WGCNA (Zhang & Horvath, 2005) (version 1.51) and coexpp (version 0.1.0, https://bitbucket.org/multiscale/coexpp) R packages. WGCNA leverages natural variance in expression between sampled individuals and timepoints to build a network structure from the Pearson correlations for all gene–gene pairs (Zhang & Horvath, 2005). WGCNA converts the gene–gene correlation matrix into an adjacency matrix using a power function that optimizes for scale-free topology, and adjacencies are then transformed into a topological overlap matrix (TOM) that represents normalized counts of neighbors that are shared between the nodes on either side of each edge. Genes were grouped using average-linkage hierarchical clustering of the TOM, followed by a dynamic cut-tree algorithm that divides the dendrogram branches into gene coexpression network modules (coEMs; Langfelder *et al*, 2008). coexpp is a specialized implementation of WGCNA that optimizes memory and multicore usage. Gene expression data were pre-processed for WGCNA by applying a $\log_2$ transformation to the FPKM quantification and removing the lowest-variance quartile of genes. Relationships among coEMs and the other data were evaluated using eigengenes (the first principal component of each coEM), calculating the Pearson correlations for all possible pairings of the coEM eigengenes, clinical variables, and cell subpopulations (Langfelder & Horvath, 2007). Network layout was performed using the ForceAtlas2 algorithm in Gephi (Bastian *et al*, 2009; version 0.9.1) followed by visualization in Cytoscape (Smoot *et al*, 2011; version 3.4.0).

## Gene set enrichment analyses

The acute-convalescent and viral titer DET signatures were analyzed for enrichment of Gene Ontology (GO) biological process (The Gene Ontology Consortium, 2015), PANTHER (Mi *et al*, 2013), and Reactome (Fabregat *et al*, 2016) terms using the Enrichr platform (Chen *et al*, 2013). DETs were selected based on varying FDR and FCH thresholds to create sets of ~1,000 DETs and mapped to unique gene symbols, which all produced qualitatively similar results for the top enriched terms; representative results for those DETs are presented in this study. Enrichr improves on the typical method of ranking term significance with one-sided Fisher's exact tests by multiplying their log-scaled *P*-values by a Z-score of the deviation from the expected rank for each term, which decreases the bias of the Fisher's exact method toward terms with few gene assignments

(Chen *et al*, 2013). Enrichment of WGCNA coEMs for terms from GO Biological Process (2015), Reactome (2016), and WikiPathways in 2016 (Kutmon *et al*, 2016) was similarly calculated using Enrichr without ranking or cutoffs. Enrichment of DET signatures within each coEM was calculated using one-sided Fisher's exact tests and a Benjamini–Hochberg adjustment.

### Statistical analyses

Analysis of paired data (Luminex, CyTOF, and RNA-seq transcript and gene-level quantifications) was performed using mixed-effects models in the limma package (version 3.30.6) as described above. Luminex analyte concentration and CyTOF cell population frequency data were log-transformed prior to analysis. For CyTOF data, the signed-rank test and Mann–Whitney $U$-test were used to compare marker intensity for paired and unpaired analyses, respectively. We used either Spearman's $\rho$ (sub-community frequencies vs. viral and antibody titers) or Pearson's $r$ (multiscale network analysis) for hypothesis testing of correlations. Visualization of small correlation matrices was performed with the corrplot R package. Quantitative measures of external cluster validity were calculated using the clusterCrit R package (version 1.2.7). $P$-values in this study were adjusted for multiple hypotheses using Benjamini–Hochberg approach, which controls the FDR (aka $q$-value). Differential expression of all genes between acute and convalescent timepoints was assessed using mixed-effects models and considered significant at an FDR threshold of < 0.05 and fold change (FCH) > 2. Elastic net regularized regression, which fits a logistic regression model with L1 and L2 penalties (the elastic net penalty), was performed with the glmnet (Friedman *et al*, 2010) R package (version 2.0-5). Elastic net hyper-parameters $\alpha$ and $\lambda$ were both selected empirically per model by a grid search that maximized AUC under fivefold nested cross-validation. 100 bootstrap resampling runs were used to estimate the 95% confidence interval for the AUC. R version 3.2.2 was used for all analyses, and in addition to those already mentioned, the following package versions were used: ggplot2 (2.2.1), pheatmap (1.0.8), ROCR (Sing *et al*, 2005) (1.0-7), and Biobase (2.30.0).

## Data and software availability

The datasets and computer code produced in this study are available in the following databases:

- RNA-seq data: Gene Expression Omnibus GSE99992 (https://www.ncbi.nlm.nih.gov/geo/query/acc.cgi?acc = GSE99992).
- Clinical data, study protocols, Luminex data, and CyTOF data: ImmPort SDY1288 (http://www.immport.org/immport-open/public/study/study/displayStudyDetail/SDY1288).

**Expanded View** for this article is available online.

### Acknowledgements

This work was supported by NIH/NIAID grants U19AI118610 (DM, TRP, AR, E-YK, SK-S, AB, MS-F, SW, AK, EH), R33AI100186 (AB, EH), and F30AI122673 (TRP), and in part by the resources and expertise of the Department of Scientific Computing at the Icahn School of Medicine at Mount Sinai. We thank Jesse Waggoner for advice regarding quantifying CHIKV viremia in patient samples. We thank study personnel at the HIMJR and the National Virology Laboratory of the Ministry of Health in Managua, Nicaragua, for enrolling patients, collecting blood samples and clinical data, maintaining databases and a high level of quality control, and preparing PBMCs. We are grateful to the study participants and their families.

## Author contributions

TRP, DM, AHR, SMW, AK, and EH designed the study. EH and AB directed studies in Nicaragua to obtain the samples for this study. AHR, DM, and SK-S performed the CyTOF experiments. AHR, SK-S, and EDA performed manual gating of CyTOF data and clustered events with MetaHybridLouvain and viSNE. AHR and EDA developed the MetaHybridLouvain algorithm. EH and DM selected study participants; DM analyzed demographic data and performed viral titer assays. DM and SK-S performed Luminex assays. E-YK, MGS, and GPT prepared RNA-seq libraries, performed sequencing, pre-processed the read data, and performed gene-level quantification. TRP performed transcript-level quantification and constructed network and predictive models. LW and JZ performed cell component analysis from gene expression and CyTOF data. MS and MS-F performed statistical analyses. TRP and DM wrote the first draft of this manuscript. All authors reviewed and approved the final version of the manuscript.

## Conflict of interest

The authors declare that they have no conflict of interest.

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
