## [Review Process File · Molecular Systems Biology]

Comprehensive innate immune profiling of chikungunya virus infection in pediatric cases

Daniela Michlmayr, Theodore R. Pak, Adeeb H. Rahman, El-Ad David Amir, Eun-Young Kim, Seunghee Kim-Schulze, Maria Suprun, Michael G. Stewart, Guajira P. Thomas, Angel Balmaseda, Li Wang, Jun Zhu, Mayte Suarez-Fariñas, Steven M. Wolinsky, Andrew Kasarskis and Eva Harris.

Review timeline:

Submission date:	3 rd July 2017
Editorial Decision:	13 th November 2017
Revision received:	9 th February 2018
Editorial Decision:	3 rd May 2018
Revision received:	31 st May 2018
Accepted:	29 th June 2018

Editor: Thomas Lemberger

Transaction Report:

1st Editorial Decision

13th November 2017

Thank you again for submitting your work to Molecular Systems Biology. I greatly apologize again for the very slow process which was due to difficulties in finding available reviewers and obtaining their reports. We have now finally heard back from the two referees who agreed to evaluate your manuscript. As you will see from the reports below, the referees find the topic of your study of potential interest. They raise, however, substantial concerns on your work, which should be convincingly addressed in a revision of this work. The constructive suggestions made by the reviewers are very clear in this regard. The issue related to potential batch effects, to the statistical analyses are particularly relevant.

Reviewer #2 also indicated the very serious efforts should be undertaken to clarify the main text and highlight better the biology.

The datasets reported in this study should be deposited in appropriate repositories (see <http://msb.embopress.org/authorguide#datadeposition>) and the relevant accession numbers or link listed in a Data and Software Availability section placed after Materials & Methods and that follows the model below:

,
Data and software availability

The datasets and computer code produced in this study are available in the following databases:

- RNA-Seq data: Gene Expression Omnibus GSE46843
[<https://www.ncbi.nlm.nih.gov/geo/query/acc.cgi?acc=GSE46843>]
- Chip-Seq data: Gene Expression Omnibus GSE46748

[<https://www.ncbi.nlm.nih.gov/geo/query/acc.cgi?acc=GSE46748>]
- Protein interaction AP-MS data: PRIDE PXD000208
[<http://www.ebi.ac.uk/pride/archive/projects/PXD000208>]
- Imaging dataset: Image Data Resource doi:10.17867/10000101
[<http://doi.org/10.17867/10000101>]
- Modeling computer scripts: GitHub
[<https://github.com/SysBioChalmers/GECKO/releases/tag/v1.0>]
- [data type]: [full name of the resource] [accession number/identifier] ([doi or URL or identifiers.org/DATABASE:ACCESSION])
,

If you feel you can satisfactorily deal with these points and those listed by the referees, you may wish to submit a revised version of your manuscript.

REFREE REPORTS.

Reviewer #1:

The manuscript by Michlmayr et al is a descriptive study to understand infection and immunity of chikungunya virus infection in pediatric patients. The study involved multiple high throughput approaches from high density mass cytometry, RNASeq to multiplex cytokine measurements from 42 children naturally infected with the virus. The study showed that during early infection, monocyte subsets are the main populations involved in early response. The study is technically strong. However, the study remains solely observational and results validates earlier publications from CD14+ monocytes and macrophages to the immune mediators signature.

Specific comments:

1. Have the authors used another anti-CHIKV antibody targeting the non-structural proteins to indicate active replication?
2. What about the comparison in children who are non-viremic? How do the subsets appear? This will be an interesting comparison as children typically clear viremia faster than in adults.
3. The CyToF data provided a wealth of information and the study will benefit if the authors could draw some association with chronic/longterm outcomes of these patients.

Reviewer #2:

Michlmayr et al. aimed to perform systematic exploration and profiling of the immune response to CHIKV by analyzing data from whole blood, PBMCs and serum of 42 pediatric cases in two time points post viral infections (acute and convalescence) by using three high multidimensional techniques of RNA-seq, CyTOF and Luminex, trying to unravel their linkage to dynamics of infection, the viral titer, the titer of anti -CHIKV virus and the clinical severity in the acute phase.

Comments:

1. The authors used CD45 based barcoding approach to pool paired samples of the same individual from two time points post infection in order minimize intra-individual potential batch effects.

Did the authors test for possible inter-individual batch effects for example as a result of different batch runs? Did they use analysis methods for batch effect removal?

2. The authors used logistic regression classifier with training data in which cell subsets were determined manually, to define major subsets in the remaining samples.

there are several concerns regarding the model assumptions:

Logistic regression requires each observation to be independent, however in the study cohort there is dependency between samples of the same individual that present different time points. In addition, the logistic regression model should have little or no multicollinearity, however some CyTOF markers may be expressed on the same cell type, therefore there might be dependency between the independent variables in the model.

- Did the authors test the logistic regression model performance?

- It is recommended to supply more details regarding the logistic regression model, at least in the supplementary regarding the markers that were included in the model and the relevant coefficients for the classification of each subset.

- The authors found several cell subsets that the CHIKV virus preferentially infect.

In the paper it was noted that one of the four sub-communities of naïve B cells also exhibits correlation with CHIKV E2 protein, but it would be good to supply more characteristic details regarding the unique marker distribution that distinguish this sub community from the other three subgroups.

5. The authors evaluate the clustering of samples according to several parameters (acute/convalescence phases, Ab titer, Viral titer, severity) by different sets of cell subset/sub-community frequency (Fig 2B, 3F) qualitatively.

In order to make quantitative estimation, external indices for cluster validity can be evaluated including entropy, purity, precision, recall or F-measure.

6. The authors did not find significant spearman correlation between CyTOF and luminex features.

a. this was followed by focused analysis of Pearson correlation between monocyte subpopulations and all cytokines. Why different correlation coefficients (Spearman Vs Pearson) were used?

b. More targeted search can be applied by looking for biological derived correlations between the cytokine and their paired receptors. In this way, it can be tested whether there is a correlation between CCL2 that was found to be highly correlated with the frequency of the monocytes subpopulations and its receptors CCR2 and CCR4.

c. Luminex analysis is often much better performed directly on MFI rather than on the concentrations - this may be something the authors can consider doing.

7. The authors found significant change in monocytes subsets using CyTOF, and a change in markers intensity that differentiate between the relevant sub-communities.

Do the relevant markers were also found to be differentially expressed in the GX data between time points? If such differences are also observed in the GX data, it might indicate that these changes were originated from the change in monocytes abundance.

8. Building a network which includes multiple data types one always runs into the issue that due to technical noise within assay results better correlate than across assays. There are multiple ways of dealing with this (e.g. different cutoffs, separate within data and across data calculations etc). Doing so may eliminate much technical based noise from data and highlight the more interesting signals. Similarly - give the authors identification of an interesting cell subset - perhaps a better approach would be to simply focus on what correlates with it.

9. Notably - gene expression data in whole blood greatly reflects cell composition, and it would be of interest to adjust the data for this.

10. I would suggest edits to the text to flesh out here the biology and compact the technical analyses, much of which should be moved to supp or Methods.

Minor comments:

- The statistical test in addition to FDR in parenthesis is not always included in the text i.e. Wilcoxon signed-rank test in line 213.

-Figure 7B: it is recommended to also add secondary y axis for q values (as shown in figure S26) or for total count of overlapping DETs, because the DET enrichment in specific module depends on the module size.

1st Revision - authors' response

9th February 2018

Point-by-point response to reviewers

MSB-17-7862: “Comprehensive innate immune profiling of chikungunya virus infection in pediatric cases”

Reviewer #1: The manuscript by Michlmayr et al is a descriptive study to understand infection and immunity of chikungunya virus infection in pediatric patients. The study involved multiple high throughput approaches from high-density mass cytometry, RNA-seq to multiplex cytokine measurements from 42 children naturally infected with the virus. The study showed that during early infection, monocyte subsets are the main populations involved in early response. The study is technically strong. However, the study remains solely observational and results validate earlier publications from CD14+ monocytes and macrophages to the immune mediators signature.

Specific comments:

1. Have the authors used another anti-CHIKV antibody targeting the non-structural proteins to indicate active replication?

We thank the reviewer for this comment and we agree that definitive conclusions about viral replication of CHIKV cannot be drawn but unfortunately, for our study we only had access to the E antibody provided by our collaborator. Due to time constraints and urgency of the study we were not able to use an antibody for the non-structural protein of CHIKV. Hence, in our study we are only able to identify CHIKV E+ cells rather than infected cells. We believe we have made this important detail sufficiently clear in the manuscript (see line 245, page 10) and hope that we can address this issue in more detail in future experiments.

2. What about the comparison in children who are non-viremic? How do the subsets appear? This will be an interesting comparison as children typically clear viremia faster than in adults.

The reviewer raises a very intriguing question and we agree that the comparison between non-viremic and viremic children would be interesting. However, the focus of our study was to characterize the immune profile in CHIKV-infected children in the acute and convalescent phase of the

disease. In this context, one of the inclusion criteria of our study was to be CHIKV+ in blood at the acute phase of infection as measured by quantitative RT-PCR, resulting in the exclusion of non-viremic children. However, we have collected acute (viremic) and convalescent (non-viremic) samples of all children and compare these two time points in our study. Within the CyTOF dataset, we included the acute-phase viral titer level as a co-variant in our heatmaps of log₁₀-scaled PBMC community frequencies for all samples and could not identify significant correlations between sub-community frequencies and log-transformed acute-phase viral titers at FDR<0.1 (Fig 2B and 3E).

3. The CyTOF data provided a wealth of information and the study will benefit if the authors could draw some association with chronic/longterm outcomes of these patients.

We agree with the reviewer that it would be highly informative and important for clinicians to correlate our datasets (CyTOF, Luminex and RNA-seq) with disease outcome. As part of our study design we collected blood samples 3, 6 and 12 months post infection and investigated any chronic or long-term effects of CHIKV in all patients. We found that none of the children in our study developed chronic arthralgia, pain and arthritis caused by CHIKV. For clarity, we have included this detail in our manuscript in the Results and Materials and Methods section (Line 162, page 7 and Line 778, page 32).

Reviewer #2: Michlmayr et al. aimed to perform systematic exploration and profiling of the immune response to CHIKV by analyzing data from whole blood, PBMCs and serum of 42 pediatric cases in two time points post viral infections (acute and convalescence) by using three high multidimensional techniques of RNA-seq, CyTOF and Luminex, trying to unravel their linkage to dynamics of infection, the viral titer, the titer of anti - CHIKV virus and the clinical severity in the acute phase.

Comments:

1. The authors used CD45 based barcoding approach to pool paired samples of the same individual from two time points post infection in order minimize intra-individual potential batch effects. Did the authors test for possible inter-individual batch effects for example as a result of different batch runs? Did they use analysis methods for batch effect removal?

We thank the reviewer for their attention to the CD45 barcoding approach and the well-placed concern about the possibility of other inter-individual batch effects that would not be controlled for by this approach. We tested for the possibility of batch effect in the final matrix of CyTOF subpopulation frequencies by looking at 3 different components: 1) acquisition, 2) staining and 3) thawing dates for each sample. Using the first principal component of the CyTOF data, which captures 78% of the variance, we evaluated the contribution of these 3 variables to the variance and found no association (all show $P > 0.7$). To quantify how much variability in the data is explained by the batch variables, we used principal variance component analysis

(PVCA). Since the 3 potential sources of batch effect were largely collinear (Spearman correlation > 0.93 for thaw, staining and acquisition batch variables), we included only the acquisition date in our PVCA analysis, which indicated that the batch explains 3.8% of the overall variance (see Figure below). This contribution is much lower than that of the “interesting” clinical variables—such as patient ID, patient age, and time point—which collectively contribute to 79.5% of the observed variance, or approximately the size of the first principal component. We believe this analysis justifies removing the “acquisition date” batch variable from downstream modeling, as the costs of increased model complexity (testing for interactions, compensating for unequal batch sizes, etc.) are unlikely to be outweighed by its minimal explanatory value. As the reviewer mentions, we did design the CyTOF experiments to minimize intra-individual potential batch effects by always pairing samples for the same individual into the same batch. Our PVCA analysis therefore indicates that inter-individual batch variability between multiple paired patient samples did not add substantially to unintended variance. In our revised manuscript, we explicitly include text in the Methods section at line 815, page 33 and in Supplemental Methods (Line 241, page 34) that summarizes the approach we used to quantify the contribution of batch effect to CyTOF data and the ensuing decision to drop the batch variable from further modeling, and added the above graphical summary as Supplemental Figure S29.

Figure S29. Principal variance component analysis of CyTOF data for evaluation of potential batch effect. *A*, Barplot of contributions for each variable to overall variance. Acquisition date explains 3.8% of the overall variance. *B*, Scatterplot of all samples in principal component space for the first two principal components. Acquisition date is used to color the samples, and grey lines connect pairs of samples across the two timepoints. As expected, the first principal component (explaining 78% of the overall variance) roughly parallels the timepoint contrast, while the acquisition date variable does not correlate with either of the first two principal components

2. The authors used logistic regression classifier with training data in which cell subsets were determined manually, to define major subsets in the remaining samples.

there are several concerns regarding the model assumptions:

Logistic regression requires each observation to be independent, however in the study cohort there is dependency between samples of the same individual that present different time points.

We thank the reviewer for their observations regarding our analysis approach. As the reviewer notes, the experimental design does indeed involve dependency between samples that came from the same individual. However, this dependency does not affect the logistic regression model, since the input for the NOD classifier is single events and their channel intensities, not samples. The classifier receives no sample-related features and does not make any sample-related predictions. For the purpose of the model, the events are independent of and unrelated to the sample structure.

-In addition, the logistic regression model should have little or no multicollinearity, however some CyTOF markers may be expressed on the same cell type, therefore there might be dependency between the independent variables in the model.

The reviewer is correct that there might be collinearity between the channel intensities in the data. This could lead to incorrect coefficient estimates, where the effect of a given channel will be overestimated while that of a correlated channel would be diminished. While this would pose a problem if we wanted an explanatory model where the coefficients are expected to provide some insight regarding the cell subsets we are predicting, this is actually not the case in this study; the goal of the model is simply prediction accuracy, not the explanatory power of the coefficients. As far as prediction accuracy is concerned, the model performs as expected.

3. Did the authors test the logistic regression model performance?

We did test the logistic model performance over the cell subsets, both by manually examining the prediction and using the F-measure (aka F1 score). Additionally, we tried using regularization (in order to address the collinearity issue raised above) and reached comparable results. These results are presented below and included in the revised manuscript in the Supplemental Methods section (Line 256, page 34) and as Supplemental Figure S30. In general, our logistic regression model shows acceptable performance for all cell subsets as measured by both precision/recall (Fig

S30A) and F1 score (Fig S30B), with larger subsets showing better performance due to the increased availability of training data.

Figure S30A. High precision and recall scores show that the classifier is returning accurate results with low false negative rates. Precision versus recall scores are plotted for each cell subset. The NOD classifier was trained over all samples except one, then applied to the remaining sample. This process was repeated for all samples ("jackknifing"). For each cell subset, precision (TP / TP + FP) and recall (TP / TP + FN) values were calculated, and the mean over all samples is presented here. (TP= true positive, FP= false positive, FN=false negative)

Figure S30B. The classifier is an accurate classifier as shown by high F1 score values for all large cell subsets. Shown is the F1 score for each cell subset. The F1 score was calculated as the harmonic mean of precision and recall ($2 \times \text{precision} \times \text{recall} / (\text{precision} + \text{recall})$). The line denotes the mean F1 score over all subsets.

-It is recommended to supply more details regarding the logistic regression model, at least in the supplementary regarding the markers that were included in the model and the relevant coefficients

for the classification of each subset.

All of the surface marker channels were used for the classifier, and we have included a table of the relevant coefficients for the classification of each subset below and in the manuscript as Table S9. However, as discussed above, we believe that the importance of these coefficients is somewhat limited since the classifier aims to build a predictive model, not an explanatory model.

Table S9. Coefficients for the logistic regression model used for classification of the major cell subsets (NodLabel).

channel	b_cell	basophil	mdc	monocyte	nk_cell	nkt_cell	pdc	t_cell
CD57	-0.116	-0.413	0.011	-0.471	0.353	0.217	-0.415	0.383
CD19	2.185	0.000	-0.933	-0.774	-1.543	-0.120	-0.680	-0.798
CD45RA	0.444	-1.009	-0.041	0.181	0.176	-0.098	0.205	0.107
CD141	-0.567	-0.434	0.173	0.072	-0.265	-0.124	0.226	0.082
CD4	-0.541	-1.087	-0.382	-0.100	-0.077	-0.091	1.010	0.105
CD8	-0.977	-0.421	-0.832	-0.574	-0.568	0.392	-0.252	0.216
CD20	1.018	0.000	-0.607	-0.351	0.000	-0.208	-0.305	-0.199
CD16	-2.190	-0.056	-1.508	1.802	0.262	-0.320	0.000	-0.950
CD127	-1.286	0.000	-1.081	-0.652	-0.702	-0.334	-0.484	0.124
CD1c	0.000	0.000	0.229	-0.392	-0.099	-0.055	-0.023	-0.049
CD123	0.252	3.691	-0.117	-0.091	-0.500	0.260	2.766	-0.550
CD66b	-0.351	-0.149	0.621	-0.472	-0.051	0.213	0.000	-0.025

CXCR5	0.369	0.000	-0.586	-0.712	0.051	0.011	-0.247	-0.285
CD86	0.343	-0.801	-0.289	0.131	-0.381	0.555	0.000	-0.205
CD27	0.078	0.000	-0.486	-0.330	-0.026	0.009	-0.372	0.194
CCR5	-0.410	0.000	0.241	-0.025	-0.654	-0.221	0.021	-0.030
CD11c	-0.155	0.000	1.749	0.307	-0.006	-0.223	-1.376	-0.452
CD14	-2.413	0.000	-2.466	3.183	-0.187	0.311	-0.068	-0.290
CD56	-0.107	-0.115	-0.889	-1.504	2.630	3.975	0.000	-4.611
CD80	-0.158	0.000	0.106	0.163	-0.200	-0.502	0.129	-0.038
CCR4	0.256	0.000	0.348	0.151	-0.031	0.126	-0.482	0.209
CD40	0.302	-0.184	0.127	-0.065	-0.063	0.144	-0.141	-0.244
CCR6	-0.307	0.000	-0.114	-0.336	0.000	0.205	0.000	0.088
CD25	0.345	0.733	-0.236	-0.630	0.297	-0.226	-0.273	0.000
CCR7	0.000	0.000	-0.254	0.124	-0.844	-0.379	0.139	-0.001
CD3	-0.602	-0.011	-0.556	-0.639	-2.269	2.809	-0.314	2.193
CX3CR1	-0.129	-0.278	0.000	0.226	0.289	-0.087	0.364	-0.602
CD38	0.165	0.040	-0.229	-0.248	0.388	-0.204	0.356	-0.150
CD161	-0.165	-0.609	-0.428	-0.726	0.512	-0.025	0.023	-0.173
CD209	-0.064	0.000	0.538	0.000	0.584	0.179	0.000	-0.181
CXCR3	-0.406	-0.685	0.000	-0.146	0.313	0.007	0.967	0.100
HLADR	0.547	-1.758	0.987	0.373	-1.134	0.114	0.659	-0.239
PD1	-0.477	0.000	-0.719	0.603	-0.021	0.160	0.000	0.350
CD54	-0.398	-0.025	-0.299	0.000	-0.165	0.021	0.196	-0.100
CD11b	-0.475	0.000	-0.406	0.043	-0.459	-0.212	-0.787	-0.587
DNA	-0.144	-0.079	0.253	0.110	-0.124	-0.022	0.000	0.235

An important overall consideration regarding the logistic classifier is that it is not intended as a tool for comprehensive analysis and classification of the data, but is instead a preprocessing step that stratifies major cell subsets prior to performing unbiased Louvain community detection, with the primary goal of improving the detection of rarer cell subsets. Thus, the classifier is only an intermediate analysis, and all the population clusters identified through the combined automated classifier and community detection analysis are ultimately manually reviewed and annotated, thereby overcoming potential inaccuracies in the initial automated classification.

4. The authors found several cell subsets that the CHIKV virus preferentially infects. In the paper it was noted that one of the four sub-communities of naïve B cells also exhibits correlation with CHIKV E2 protein, but it would be good to supply more characteristic details regarding the unique marker distribution that distinguish this sub community from the other three subgroups.

We appreciate the reviewer's interest in the sub-community of naïve B cells whose frequency significantly correlated with CHIKV E2 protein expression, and agree that this might also be of interest to the general readership. To characterize the details of how this sub-community's marker expression differs from the other naïve B cells, we used a similar analysis and heatmap visualization as in Figure S7, which depicted the significantly different channels among the CD14+ monocytes. We have added the new analysis for the four sub-communities of naïve B cells in a new Supplementary Figure S2, which is reproduced below. This analysis is restricted to markers that differentiate sub-community 4 from the median value of the other three sub-communities at the threshold of a fold change > 1.5 and FDR < 0.05 (using a modified paired t-test under the mixed effects model), with colors representing \log_2 fold change in channel intensity from the median for all communities (red denotes higher intensity). Notably, sub-community 4 is

characterized by a much higher expression of CXCR5 as compared to all other sub-communities of naïve B cells. This chemokine receptor, together with CCR7, is important for the migration of lymphocytes to secondary lymphoid tissues and is implicated in follicle formation within tissues.^{1,2} Otherwise, the pattern of marker expression of sub-community 4 is very similar to sub-community 1 of naïve B cells, albeit with much lower expression of CCR4, CXCR3 and CD80. CXCR3 has been described as a marker for malignant B cells and is absent in normal naïve B cells in the blood.³ CD80 is a marker that is pivotal for the activation of autoreactive T cells and plays an important role in the development of a form of rheumatoid arthritis.⁴ In the light of this literature, it appears that the different types of naïve B cell communities that we detect in CHIKV-infected children reflect different degrees of activation. Subcommunity 1 and 4 seem to be more activated based on the higher expression of CCR4, CXCR3 and CD80 and could be transitioning into memory B cells.

Given these interesting findings, we have added a reference to this analysis in the Results on line 243, page 10, and included additional discussion about the above findings in the Discussion on line 584, page 24.

Figure S2. Heatmap of differences in marker protein expression between four naïve B cell sub-communities identified by MetaHybridLouvain. All channels listed here differentiate sub-community 4 from the median value for the other three sub-communities at a threshold of fold change > 1.5 and $FDR < 0.05$ (using a moderated paired t-test under the mixed effects model). Colors represent \log_2 fold change in channel intensity from the median for all communities (red means higher intensity).

5. The authors evaluate the clustering of samples according to several parameters (acute/convalescence phases, Ab titer, Viral titer, severity) by different sets of cell subset/sub-community frequency (Fig 2B, 3F) qualitatively. In order to make quantitative estimation, external indices for cluster validity can be evaluated including entropy, purity, precision, recall or F-measure.

We agree with the reviewer that a quantitative measure of cluster validity would enhance the manuscript's qualitative evaluation of how well

hierarchical clustering separates CyTOF sub-community frequencies into acute vs. convalescent signatures. As suggested, we have calculated such external indices for the top-level clustering (incorporating MetaHybridLouvain sub-community frequencies) presented in Figure 3F, which produces: Jaccard similarity coefficient = 0.87, precision = 0.93, recall = 0.93, and F-measure = 0.93. This confirms the two major takeaways for the qualitative assessment we originally presented, which was that (1) clustering of MetaHybridLouvain sub-community frequencies separates the samples by time point with high accuracy, and that (2) performance is higher when using MetaHybridLouvain's unbiased identification of all leukocyte sub-communities as opposed to the canonical population labels alone (by comparison to Figure 2B, where Jaccard index = 0.61, precision = 0.76, recall = 0.76, and F-measure = 0.76). We thank the reviewer for this suggestion, as we believe this analysis enhances the manuscript, and we now report the above statistics in the Results section (lines 201 and 254 on pages 9 and 11, respectively). We have also added to the Materials and Methods that the tests were performed with the 'clusterCrit' R package (version 1.2.7) on Line 932, page 38.

6. The authors did not find significant spearman correlation between CyTOF and luminex features.

a. this was followed by focused analysis of Pearson correlation between monocyte subpopulations and all cytokines. Why different correlation coefficients (Spearman Vs Pearson) were used?

We appreciate the reviewer's concern that these analyses were performed with different correlation coefficients, which unnecessarily complicate comparisons across the series of figures, and we apologize for this technical oversight. In the revised manuscript, we have replaced the analysis in Figure S16 with Spearman's correlations, which results in a consistent method for Figures S13-S16, and we have updated the corresponding paragraph in the Results. This leaves our overall interpretation unchanged, as the magnitude of the correlations in Fig S16 was slightly diminished by using Spearman's correlation, but the convalescent-phase correlations for CCL2 are still significantly different from the correlations against all other cytokines at a new FDR of $P = 0.015$ (this is updated in the Results section on Line 343, page 14). In this new analysis, CCL2 is the only cytokine to show significantly different correlations at a FDR P threshold of <0.05 . Therefore, we retain the original conclusion of this section—that CCL2 appears to have a comparatively important regulatory role for monocyte populations during CHIKV infection progression. The new Fig S16 is reproduced below.

Figure S16. Clustered heatmap of Pearson correlations between log-scaled serum cytokine concentration and log-scaled monocyte subphenotype frequencies. *A*, within acute-phase samples. *B*, within convalescent-phase samples. CCL2 within the convalescent timepoint (highlighted) displayed the only set of Pearson correlations that differed significantly from those of the other cytokines at an FDR threshold of < 0.05 (Mann-Whitney U test).

b. More targeted search can be applied by looking for biological derived correlations between the cytokine and their paired receptors. In this way, it can be tested whether there is a correlation between CCL2 that was found to be highly correlated with the frequency of the monocytes subpopulations and its receptors CCR2 and CCR4.

We thank the reviewer for this valuable suggestion to apply a targeted search for chemokines and their cognate receptor. Since most chemokine receptors are not soluble in circulation, they cannot be measured by Luminex only. Therefore we have compared the protein levels of the chemokines with the gene expression levels of the corresponding chemokine receptor using Spearman correlations. We found that during the acute phase, the expression of CCR2 is significantly correlated with levels of CCL2 protein (FDR $P < 0.05$). CCR2 is a receptor predominantly expressed by monocytes, while CCL2 can be secreted from various different cell types. This high correlation during the acute phase aligns with our CyTOF results in that CCR2 is mainly expressed on monocytes, and we find that monocytes are expanded during the acute phase. While CCR4 is another ligand for CCL2, we did not find a significant correlation when comparing ligand and receptor. However, notably, CCR4 is typically expressed by T cells and not monocytes. Clearly, CCR2 is highly expressed

during the acute phase of infection in concurrence with CCL2's significant upregulation at this timepoint (Fig 5B), which is consistent with the monocyte-centric response detailed by our study.

Figure Legend: Correlation plots of chemokine ligands and their cognate chemokine receptor during the acute and convalescent phase of CHIKV infection. Spearman's correlations between the protein concentrations of the cytokines profiled by Luminex and the abundance of their receptors, quantified by RNA-seq at the gene expression level for acute and convalescent (conv) time points. The overall changes of ligand and receptor levels comparing convalescent versus acute. (+ $p < 0.1$; * $p < 0.05$; ** $p < 0.01$; *** $p < 0.001$)

c. Luminex analysis is often much better performed directly on MFI rather than on the concentrations - this may be something the authors can consider doing.

The reviewer correctly notes that analysis directly on MFI values can provide benefits over using concentration values because it obviates censoring of values beyond the detection limits, and this may increase power.⁵ Therefore, we have also analyzed the data using MFI values and find that both methods identify the same set of cytokines to be significantly different—with the sole exception of TGF- α , which is significant at FDR $P < 0.05$ using the MFI data, but for which the FDR-adjusted P value for the standard-curve concentrations is 0.06. Moreover, a high correlation is observed between the calculated fold changes for each method (Pearson's $r = 0.86$, 95% CI [0.75–0.92]; Spearman's $\rho = 0.93$, 95% CI [0.84–0.96]). Since our overall results are essentially unchanged under either approach, we have opted to present the analysis using concentrations (in pg/mL) as it is more readily interpretable to a biological audience and simpler to compare to prior studies of CHIKV-induced cytokine changes, which uniformly present their Luminex results using standard-curve concentrations.^{6–10}

The Figure below is a scatterplot of log fold changes estimated from net

MFI values against log fold changes estimated from concentration values for all cytokines; the high correlation previously mentioned is readily observed as the positive linear trend, and the one exception to statistical significance as determined by the FDR-adjusted P value threshold of 0.05 is labeled (TGfA). In general, the two analyses result in concordant significantly (red) and non-significantly (blue) differentially expressed cytokines.

Figure Legend. Protein concentration changes and net mean fluorescence intensity (MFI) values are correlated. Log fold changes as determined by analysis of concentration values (X axis) against the same as determined by net mean fluorescence intensity (MFI) values (Y axis). As expected, a positive correlation is observed (Pearson's $r = 0.86$, Spearman's $\rho = 0.93$). Each cytokine is represented as a separate point, and is colored by whether it reaches statistical significance at FDR-adjusted $P < 0.05$ (Mann-Whitney U) in neither, one, or both analyses.

7. The authors found significant change in monocytes subsets using CyTOF, and a change in markers intensity that differentiate between the relevant sub-communities. Do the relevant markers were also found to be differentially expressed in the GX data between time points? If such differences are also observed in the GX data, it might indicate that these changes were originated from the change in monocytes abundance.

We thank the reviewer for this interesting idea. It prompts a focused query of gene expression changes for the CyTOF markers differentiating the various subpopulations of monocytes, with the hypothesis that the observed subpopulation phenotypes and frequency shifts may help explain the gene expression changes. Given the interesting sub-phenotypes we discovered for $CD14^+$ monocytes in this study, including one (sub-community 3) that displayed substantial expression of CCR4, CXCR3, and CCR6, which are markers not classically associated with monocytes, we focus our response on $CD14^+$ monocytes. Querying for differentially expressed transcripts mapping to the genes for the markers seen in Figure 4B (which are the markers that most differentiate the three sub-communities at FDR < 0.05

and fold change > 1.3) produces the table below, which is a filtered subset of Table S4. We found that CCR7, CCR4 and CCR6, which are mostly expressed on CD14⁺ sub-community 3, were significantly downregulated during the acute phase as compared to convalescence, while CD80, CD40, CD123, CD86 and CD54 (highly expressed on CD14⁺ sub-community 1) were upregulated during the acute phase of infection (both FDR<0.05). These findings from the gene expression data reflect the cell frequency changes we measured by CyTOF whereby sub-community 1 is very high during the acute phase as compared to a lower abundance of sub-community 3. Thus, our gene expression data is consistent with the CyTOF data, which indicates that these changes in gene expression originated from the change in monocyte frequency. We have included these data in the Results section in Line 310, page 13 and in Table S1.

Table S1. Gene expression analysis of markers that are significantly different between CD14⁺ monocyte subpopulations (from Fig. 4B) during acute and convalescence

Acute – Conv		lgFCH	FCH	pvals	fdrs
CCR7	CCR7	-1.1	-2.14	1.2E-11	6.52E-11
CCR4	CCR4	-0.74	-1.67	8.23E-15	6.98E-14
CCR6	CCR6	-0.73	-1.66	8.55E-10	3.6E-09
CD66b	CEACAM8	0.02	1.01	0.927	0.941
CD80	CD80	0.16	1.11	0.0207	0.0305
CX3CR1	CX3CR1	0.22	1.17	0.0515	0.0706
CD40	CD40	0.24	1.18	0.0307	0.0439
CD123	IL3RA	0.49	1.41	0.000185	0.000378
CD86	CD86	0.55	1.47	1.62E-10	7.56E-10
CD54	ICAM1	1.25	2.38	1.16E-12	7.23E-12

8. Building a network, which includes multiple data types one always runs into the issue that due to technical noise within assay results better correlate than across assays. There are multiple ways of dealing with this (e.g. different cutoffs, separate within data and across data calculations etc). Doing so may eliminate much technical based noise from data and highlight the more interesting signals. Similarly - give the authors identification of an interesting cell subset - perhaps a better approach would be to simply focus on what correlates with it.

We certainly agree with the reviewer's observation that within-assay results tend to correlate better than across assays, as this is strongly reflected in our results; in fact, the Luminex data intracorrelated to such a degree that it was eventually dropped from our cross-assay (i.e., multiscale) analyses (see Fig S27; Fig 7D; Fig S28C) for lack of predictive value compared to the CyTOF and RNA-seq data. The unsupervised clustering techniques intrinsic to the WGCNA and MultiHybridLouvain methods are actually the foundation of our strategy for reducing technical noise: they condense thousands of dimensions into simpler modules (coEMs) and sub-communities, producing more stable averages from noisy, intracorrelated input data. We prefer these clustering techniques for network construction because they increase

the “signal” of globally significant changes without involving the subjectivity of picking cutoffs/filters for the data or comparisons thereof—which may unintentionally introduce the experimenters’ bias into the signal that remains. Although we did identify an interesting cell subset (monocytes) using CyTOF toward the beginning of the study and this indeed sets the overall theme for our discussion, we intentionally tried to avoid restricting our hypotheses based on early results from one experimental method. As presented in the title, our aim is to deliver “comprehensive innate immune profiling” of CHIKV. Therefore, we place RNA-seq, CyTOF, and Luminex data on an equal footing when modeling the various clinical parameters of CHIKV infection, and this is why our manuscript presents and explores each data type *de novo* in their own sections before introducing the final integrative analysis. Furthermore, this is our rationale for equally weighting the WGCNA modules, MetaHybridLouvain communities, and clinical variables when constructing the integrative network presented in Fig 7D.

9. Notably - gene expression data in whole blood greatly reflects cell composition, and it would be of interest to adjust the data for this.

We agree with the reviewer that the main signal for gene differential expression is likely reflective of the changes in cell composition well established by the CyTOF results. In order to investigate this, from our quantified RNA-seq data, we derived lists of genes upregulated in the acute phase of CHIKV infection compared to the convalescent phase (adjusted P value <0.001 , paired t-test), as well as genes upregulated in the convalescent phase. When comparing the expression levels of these genes across a panel of hematopoietic cells of different lineages,¹¹ we observed genes with higher expression in the acute phase tend to be overexpressed in granulocytes and monocytes (panel A below; note columns corresponding to the black GM label), while genes with higher expression in the convalescent phase tend to be overexpressed in T cells and B cells (panel B; note columns corresponding to the purple and pink BCELL and TCELL labels).

Figure S24. Heatmap of gene expression across different types of blood cells for genes expressed higher in acute phase (A) or higher in convalescent phase (B). Rows represent genes. Columns represent blood cells, which are grouped according to the lineage (column legend). HSC, Hematopoietic stem cell; MYP, Myeloid progenitor; ERY, Erythroid cell; MEGA, Megakaryocyte; GM, Granulocyte/monocyte; EOS, Eosinophil, BASO, Basophil; DEND, Dendritic cell

Next, we used computational methods to estimate cell components based on gene expression profiles of admixtures. To deconvolute gene expression data to the cell type proportion, we utilized known cell subset-specific marker genes of six major blood cell types obtained from the *IRIS* project,¹² and estimated abundance of each cell type by the average expression of its markers. Also, we used the recently developed algorithm *CIBERSORT*¹³ which requires an input matrix of reference gene expression signatures of different cell subsets. The *CIBERSORT* R package and its associated leukocyte signature matrix of 22 cell types were utilized with all default parameters.

We then compared the above cell component results derived from gene expression with CyTOF results. As shown in the figure below, the computational estimation and CyTOF readouts agree well with each other: the correlation between the same or similar cell types is generally much higher (red cells; compare labels in rows vs. columns) than between different cell types (blue cells). This demonstrates an overall consistency between the gene expression and CyTOF data for the purpose of estimating cell subpopulations.

Figure S25. Correlation between cell frequency derived from CyTOF and that estimated from gene expression using expression of cell-specific markers (left) or the *CIBERSORT* algorithm (right).

Finally, as requested by the reviewer, we aimed to identify differentially expressed genes that are not purely caused by cellular component changes. To achieve this, we adjusted each gene's expression according to cellular abundance. Due to the relatively small sample size, we only considered the abundance of six main types of blood cells estimated by the average expression of cell markers (as described above): i.e., B cells, T cells, NK cells, monocytes, neutrophils and dendritic cells. Specifically, we used linear regression to obtain the residual gene expression after considering the cell abundance of the six major cell types in the model. We then used paired t tests to obtain differentially expressed genes between the acute and convalescent phases. In this way, we obtained only 132 differentially expressed genes with a nominal P value < 0.05 , and none of them remained significant at $FDR < 0.05$ after multiple hypothesis correction. Therefore, the main signal of differential gene expression in whole blood is indeed strongly derived from the changes in cell subpopulation frequencies.

We have included these new findings in the Results in lines 515, page 17, along with the new Figures S24 and S25 that are reproduced above. The technical approach for this analysis was added to the Supplementary Methods section on lines 298, page 36.

10. I would suggest edits to the text to flesh out here the biology and compact the technical analyses, much of which should be moved to supp or Methods.

We thank the reviewer for this valuable comment. We have moved many of the technical sections within the Results and Discussion to the Materials and Methods and Supplemental Methods section to make the manuscript more compact and improve readability. Furthermore, we have also expanded the biological interpretation within the analyses throughout the manuscript.

Minor comments:

- The statistical test in addition to FDR in parenthesis is not always included in the text i.e. Wilcoxon signed-rank test in line 213.

We thank the reviewer for pointing this out. We have made the manuscript more consistent and added the statistical test used for each analysis in the Materials and Methods section and also within the parentheses in the text as needed.

- Figure 7B: it is recommended to also add secondary y axis for q values (as shown in figure S26) or for total count of overlapping DETs, because the DET enrichment in specific module depends on the module size.

We agree with the reviewer that the addition of a secondary axis would be beneficial for the interpretation of Fig 7B. We have therefore added the secondary axis to Fig 7B and removed the former Fig S26 from the Supplementary Material. The new Fig 7B is depicted below, wherein the q values are depicted as red asterisks on the secondary (log-scaled) axis, now labeled in red on the right side of the plot. To aid in interpretation, the dotted horizontal line originally in Fig S26 at the traditional cutoff of $FDR < 0.05$ is added to Fig 7B as well, also in red.

Figure 7B. Enrichment of five subsets of the DET signatures for CHIKV infection phase and viral titer (see Figs 6A and 6C) among each of the 92 coEMs (X axis), showing the fractional overlap of the module with the DET signature (Y axis). Negative log₁₀ q values (Benjamini-Hochberg adjusted P values; Fisher's exact test) are depicted as red asterisks on the secondary Y axis (right hand side), clipped to a maximum of 10. The only modules with overlaps in any of the subsets that exceed $FDR < 0.05$ are turquoise and sienna. The horizontal line indicates a threshold of $FDR < 0.05$.

References

- 1 Stein J V., Nombela-Arrieta C. Chemokine control of lymphocyte trafficking: A general overview. *Immunology* 2005; **116**: 1–12.
- 2 Cozine CL, Wolniak KL, Waldschmidt TJ. The primary germinal center response in mice. *Curr Opin Immunol* 2005; **17**: 298–302.
- 3 Trentin L, Agostini C, Facco M, Piazza F, Perin A, Siviero M, Gurrieri C, Galvan S, Adami F, Zambello R, Semenzato G. The chemokine receptor CXCR3 is expressed on malignant B cells and mediates chemotaxis. *J Clin Invest* 1999; **104**: 115–21.
- 4 O'Neill SK, Cao Y, Hamel KM, Doodes PD, Hutas G, Finnegan A. Expression of CD80/86 on B Cells Is Essential for Autoreactive T Cell Activation and the Development of Arthritis. *J Immunol* 2007; **179**: 5109–16.

- 5 Breen EJ, Tan W, Khan A. The Statistical Value of Raw Fluorescence Signal in Luminex xMAP Based Multiplex Immunoassays. *Sci Rep* 2016; **6**: 1–13.
- 6 Teng T-S, Kam Y-W, Lee B, Hapuarachchi HC, Wimal A, Ng L-C, Ng LFP. A Systematic Meta-analysis of Immune Signatures in Patients With Acute Chikungunya Virus Infection. *J Infect Dis* 2015; **211**: 1925–35.
- 7 Chaaitanya IK, Muruganandam N, Sundaram SG, Kawalekar O, Sugunan AP, Manimunda SP, Ghosal SR, Muthumani K, Vijayachari P. Role of proinflammatory cytokines and chemokines in chronic arthropathy in CHIKV infection. *Viral Immunol* 2011; **24**: 265–71.
- 8 Ng LFP, Chow A, Sun Y-J, Kwek DJC, Lim P-L, Dimatatac F, Ng L-C, Ooi E-E, Choo K-H, Her Z, Kourilsky P, Leo Y-S. IL-1beta, IL-6, and RANTES as biomarkers of Chikungunya severity. *PLoS One* 2009; **4**: e4261.
- 9 Chow A, Her Z, Ong EKS, Chen J, Dimatatac F, Kwek DJC, Barkham T, Yang H, Rénia L, Leo Y-S, Ng LFP. Persistent arthralgia induced by Chikungunya virus infection is associated with interleukin-6 and granulocyte macrophage colony-stimulating factor. *J Infect Dis* 2011; **203**: 149–57.
- 10 Schilte C, Staikovsky F, Couderc T, Madec Y, Carpentier F, Kassab S, Albert ML, Lecuit M, Michault A. Chikungunya Virus-associated Long-term Arthralgia: A 36-month Prospective Longitudinal Study. *PLoS Negl Trop Dis* 2013; **7**. DOI:10.1371/journal.pntd.0002137.
- 11 Novershtern N, Subramanian A, Lawton LN, Mak RH, Haining WN, McConkey ME, Habib N, Yosef N, Chang CY, Shay T, Frampton GM, Drake ACB, Leskov I, Nilsson B, Preffer F, Dombkowski D, Evans JW, Liefeld T, Smutko JS, Chen J, Friedman N, Young RA, Golub TR, Regev A, Ebert BL. Densely interconnected transcriptional circuits control cell states in human hematopoiesis. *Cell* 2011; **144**: 296–309.
- 12 Abbas AR, Baldwin D, Ma Y, Ouyang W, Gurney A, Martin F, Fong S, van Lookeren Campagne M, Godowski P, Williams PM, Chan AC, Clark HF. Immune response in silico (IRIS): Immune-specific genes identified from a compendium of microarray expression data. *Genes Immun* 2005; **6**: 319–31.
- 13 Newman AM, Liu CL, Green MR, Gentles AJ, Feng W, Xu Y, Hoang CD, Diehn M, Alizadeh AA. Robust enumeration of cell subsets from tissue expression profiles. *Nat Methods* 2015; **12**: 453–7.

2nd Editorial Decision

3rd May 2018

Thank you again for submitting your work to Molecular Systems Biology. We have now finally heard back from the referees who accepted to evaluate the study. They are now supportive and I am pleased to inform you that we will be able to accept your paper for publication in Molecular Systems Biology pending the minor amendments listed below.

Corresponding Author Name: Eva Harris

Manuscript Number: MSB-17-7862